# GENERALIZABLE AND CONSISTENT GRANULAR EDGE PREDICTION

## ABSTRACT

We introduce a new task in edge detection: Granular Edge Prediction. Unlike traditional binary edge maps, this task aims to predict a categorical edge map, where each edge pixel is assigned a granularity level reflecting the likelihood of being recognized as an edge by a human annotator. Our contributions are threefold: 1) we construct a large-scale synthetic dataset for granular edge prediction, where each edge is labeled with a quantized granularity level, and introduce a graph-based edge representation to enforce consistency in edge granularity across the dataset, 2) we develop a novel edge consensus loss to enforce granularity consistency within individual edges, and 3) we propose a comprehensive evaluation framework, including granularity-aware edge evaluation and two quantitative metrics to assess the consistency of granular edge prediction. Extensive experiments demonstrate that our method generalizes well in zero-shot evaluation across four standard edge detection datasets, closely aligns with human perception of edge granularity, and ensures high consistency in edge-wise granularity estimation.

## 1 INTRODUCTION

Edge detection aims to identify the salient boundaries in images. It serves as a fundamental problem in computer vision and finds applications in various domains, including medical image analysis (Abdel-Gawad et al., 2020), autonomous driving (Bertozzi & Broggi, 1998), object detection (Ullman & Basri, 1989; Ferrari et al., 2007), conditional image generation (Zhang et al., 2023), and 3D curve reconstruction (Ye et al., 2023). Despite its fundamental importance, edge detection is inherently subjective. Different annotators may perceive edges differently, leading to varying levels of detail in annotations.

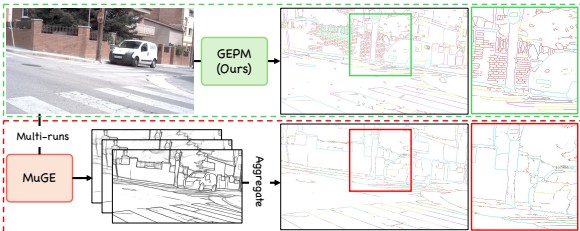

Figure 1: Existing methods like MuGE (Zhou et al., 2024) require multiple inferences for a granular prediction, while GEPM produces a single-pass prediction with more consistent edges.

Motivated by this observation, we propose Granular Edge Prediction (GEP), a task that not only detects edges but also assigns a granularity score to each edge candidate. This score quantifies the likelihood of an edge being perceived and annotated by different observers, offering a meaningful measure of edge saliency. It helps users interpret edge perceptibility, making it particularly valuable and has potential for applications where edge prominence varies, such as depth estimation(Xian et al., 2020), scene understanding(Arbelaez et al., 2010a), and artistic rendering(Simo-Serra et al., 2018). The multi-granularity nature of GEP also allows it to seamlessly adapt to tasks requiring different levels of detail. This adaptability is particularly advantageous in two scenarios: 1) tasks that demand flexible detail control, such as industrial inspection, where both subtle and prominent defects can be detected at different granularity levels, and 2) new domains where no human-labeled edges are available, enabling GEPM to provide zero-shot predictions that users can threshold based on their desired edge abstraction.

The GEP problem faces two key challenges from the scarcity of datasets and the limitations of edge prediction approaches. 1) Only two datasets, BSDS (Arbelaez et al., 2010b) and Multicue (Mély

et al., 2016), provide granular edge annotations, containing just 500 and 100 labeled images, respectively. The small dataset size leads to a limited image distribution, causing models trained on one dataset to overfit and generalize poorly to different edge distributions. This hinders real-world applications where edge statistics differ from those seen during training. 2) Traditional edge detection methods (Canny, 1986; Xie & Tu, 2015; Su et al., 2021; Liu et al., 2017; Pu et al., 2022) predict binary edge maps without granularity information. Recent approaches such as UAED (Zhou et al., 2023) and MuGE (Zhou et al., 2024) attempt to incorporate granularity into model training but suffer from two major limitations. First, as shown in Figure 1, they require multiple predictions per image to infer granularity, as a single prediction only determines whether an edge exists at a given threshold rather than its intrinsic granularity. Second, they often assign different granularity values to pixels within the same edge, contradicting human annotations where an edge is either labeled or not, implying a consistent granularity across its structure.

To address these challenges, we introduce the Synthetic Granular Edge Dataset (SGED), a large-scale dataset containing 376,515 images. Each sample consists of an RGB image paired with a synthetic edge map, where edges are annotated with 36 continuous granularity values. To ensure that the granularity pattern aligns with human annotation behavior – where an edge is either labeled or not, and its granularity remains consistent – we propose a novel graph-based edge representation to refine the synthetic edge map and enforce granularity consistency across individual edges.

For granularity-aware edge prediction, we propose the Granular Edge Prediction Model (GEPM), which simultaneously predicts edges and their corresponding granularity. Additionally, we introduce a novel Edge Consensus Loss to enhance granularity consistency within each edge by minimizing prediction divergence among its pixels. Trained on SGED, GEPM generalizes well to four standard benchmark datasets under a zero-shot testing setting. Empirical results demonstrate that GEPM maintains high edge granularity consistency under two newly proposed edge consistency metrics and aligns well with human annotations in a granularity-aware edge evaluation.

Our contributions are summarized as follows: 1) We construct a large-scale Synthetic Granular Edge Dataset, refined using a novel graph-based edge representation, ensuring granularity consistency across individual edges and enabling state-of-the-art zero-shot granular edge detection performance. 2) We develop the Granular Edge Prediction Model, which simultaneously predicts edges and their corresponding granularity. To enforce granularity consistency within individual edges, we introduce a novel Edge Consensus Loss, minimizing prediction divergence among edge pixels. 3) We provide two new edge consistency metrics to quantitatively assess granularity consistency and introduce granularity-aware edge evaluation to measure the alignment between predicted granularity and human annotations.

## 2 Synthetic Granular Edge Dataset (SGED)

### 2.1 Synthetic Edge Generation

We use images from the web-crawled LAION dataset (Schuhmann et al., 2022) as the source for our granular edge dataset. The rough idea for generating synthetic edge maps follows (Gupta et al., 2013), where semantic mask boundaries are used to construct edge maps. However, directly replicating this method presents two challenges: 1) LAION images do not come with semantic category annotations, and 2) relying solely on semantic mask boundaries results in a single binary edge map, which does not reflect edge granularity.

To address these limitations, we seek an alternative that accurately captures object contours while allowing for granularity adjustment. We achieve this using the Segment Anything Model (SAM) (Kirillov et al., 2023), which automatically detects objects in images and generates corresponding masks. We synthesize edge maps from these object masks by applying morphological erosion and computing the difference between the original and eroded masks, effectively extracting contour-like edges. By adjusting SAM's hyperparameters, we control the criteria for object recognition, which in turn influences the saliency of the extracted edges.

By iterating over all objects in an image, we generate a binary edge map for each specific configuration. To ensure diversity in edge granularity, we carefully select 36 distinct SAM configurations that produce decent variations in edge maps. Details of these configurations are provided in Section E.

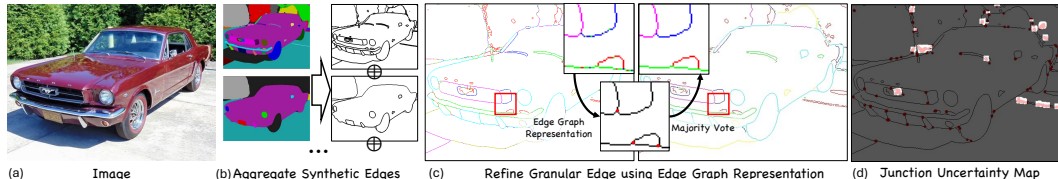

(a) Image  (b) Aggregate Synthetic Edges  (c) Refine Granular Edge using Edge Graph Representation  (d) Junction Uncertainty Map

Figure 2: GED generation pipeline aggregates synthetic edges and applies refinement by converting edges into a novel graph representation. The junction uncertainty map highlights regions with excessive junctions where gradients are blocked during training.

The final granular edge map is obtained by merging all 36 generated edge maps through pixel-wise summation, resulting in a granularity scale ranging from 0 to 36.

## 2.2 A Graph Representation for Refinement

The initial granular edge map exhibits two issues: 1) edge ambiguity, where unclear object boundaries result in thick or diffused edges rather than distinct contours, and 2) granularity inconsistency, where pixels along the same edge may have varying granularity values, contradicting human annotation patterns.

To refine edge granularity and ensure consistency, we introduce a graph-based approach that transforms the edge map into a structured graph representation. We first resolve edge ambiguity by applying Non-Maximum Suppression (NMS) with edge thinning algorithm (Guo & Hall, 1989), ensuring that edges are reduced to single-pixel contours. We then convert the thinned edge map into a connected graph by treating edge pixels as nodes and their connectivity as edges. Nodes are classified based on their degree: those with three or more degrees are junctions, those with exactly one degree are endpoints, and all others are trivial nodes. It is easy to realize that an edge in the graph is a continuous path between two junctions or endpoints, with all intermediate pixels being trivial nodes, or a loop with trivial nodes only. To enforce granularity consistency, we assign a single granularity value to each edge by selecting the most frequently occurring granularity along its path. Additionally, to reduce noise, we apply connected component analysis and remove isolated edges smaller than eight pixels, which are empirically found to be artifacts. The pipeline of edge generation and refinement is illustrated in Figure 2 and SGED samples can be found in Section G.

# 3 Granular Edge Prediction Model

## 3.1 Reformulate Granular Edge Prediction

There are two possible training strategies for utilizing the SGED dataset to achieve granular edge prediction. A straightforward approach, similar to MuGE (Zhou et al., 2024), involves setting a granularity threshold and training the model to predict only edges with granularity below the threshold. However, this method produces a binary prediction and requires multiple forward passes to determine the granularity of each edge, significantly reducing efficiency.

Instead, we challenge the deep-rooted idea that edge prediction is merely a binary classification problem and reformulate granular edge prediction as a multi-class classification problem. This reformulation transforms the task into a pixel-wise categorical prediction problem, where the total number of classes $|g|$ corresponds to the number of granularity levels plus an additional background class. Consequently, we propose the Granular Edge Prediction Model (GEPM), which distinguishes itself by predicting both edges and their corresponding granularity in a single inference step as shown in Figure 3. GEPM is trained with two losses: i) a junction confidence-aware categorical cross-entropy loss for granular prediction and ii) a novel edge consensus loss to encourage pixels belonging to the same edge to have consistent granularity.

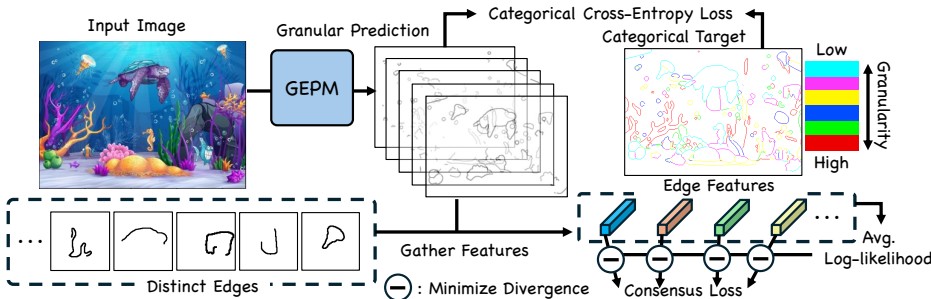

Figure 3: Our Granular Edge Prediction Model predicts pixel-wise granular probability distribution, treats edge granularity as distinct classes and is optimized with cross-entropy loss and consensus loss. The consensus loss enforces edge granularity consistency by minimizing the discrepancy between each pixel's granularity distribution and the average distribution of its corresponding edge.

We first replace the classic binary cross-entropy loss in edge detection to a pixel-wise categorical cross-entropy loss:

$$l_{\text{entropy}} = \mathbb{E}_{\mathcal{I} \sim \mathcal{D}} \sum_{i=1}^{|\mathcal{I}|} \sum_{g} -\lambda_g \mathbb{I}[y_i = g] \ln P(g), \tag{1}$$

where $y_i$ denotes the ground-truth granularity label for the $i$-th pixel in an image $\mathcal{I}$ sampled from dataset $\mathcal{D}$, and $P(g)$ represents the predicted probability of the pixel belonging to granularity level $g$. The indicator function $\mathbb{I}$ is nonzero only when the predicted granularity level matches the ground-truth label $y_i$. Following existing works (Xie & Tu, 2015; Zhou et al., 2023; 2024), we introduce a weighting factor $\lambda_g$ to balance the contributions of edge ($Y_{g>0}$) and background ($Y_{g=0}$) pixels:

$$\lambda_g = \begin{cases} |Y_{g>0}|/(|Y_{g>0}| + |Y_{g=0}|), & \text{if } g = 0, \\ |Y_{g=0}|/(|Y_{g>0}| + |Y_{g=0}|), & \text{otherwise.} \end{cases} \tag{2}$$

## 3.2 JUNCTION CONFIDENCE-AWARE TRAINING

The SGED dataset is generated automatically, which inevitably introduces noise in the annotations. While our dataset processing pipeline effectively removes most false positive edges and ensures granularity consistency, certain edges remain ambiguous due to unclear object boundaries. In such case, although the non-maximum suppression operation prevents overlapping edges, the resulting thinned edge map may still fail to accurately reflect the true boundary location. Empirically, we observe that in regions with ambiguous edges, the thinning process tends to produce an excessive number of junctions, as illustrated in Figure 2 (d).

Such ambiguity in edge annotations can lead to suboptimal model behavior during training. To address this issue, we propose a simple yet effective strategy where the model intentionally ignores regions with an excessive number of junction points. This operation is efficiently implemented by applying an all-ones convolutional kernel to the binary junction map and masking all regions where the resulting pixel value exceeds one. We refer to this approach as Junction Confidence-Aware training, which modifies the standard cross-entropy loss as follows:

$$l_{\text{entropy}}^{\text{JuncConf}} = \mathbb{E}_{\mathcal{I} \sim \mathcal{D}} \sum_{i=1}^{|\mathcal{I}|} (1 - \mathbb{I}[\text{JuncCnt}(i, r) > 1]) \, l_i, \tag{3}$$

where $\text{JuncCnt}(i, r) > 1$ computes the number of junction points surrounding the $i$-th pixel within a predefined range $r$, and $l_i = \sum_g -\lambda_g \mathbb{I}[y_i = g] \ln P(g)$ represents the cross-entropy loss for the $i$-th pixel. We empirically set $r = 3$ according to visual inspection.

## 3.3 EDGE CONSENSUS LOSS

Granularity reflects the probability that an edge will be recognized. Ideally, all pixels belonging to the same edge should be assigned the same granularity. However, in practice, we observe significant

variations in granularity predictions across individual edge pixels. A closer examination of standard edge detection training losses reveals that this inconsistency stems from the pixel-wise nature of the loss function. Models trained in this manner lack awareness of the holistic structure of edges, leading to inconsistent feature representations for pixels belonging to the same edge.

To address this issue, we introduce an edge-wise consensus loss that encourages the model to learn consistent representations for each edge. The key idea is to enforce similarity among pixels within the same edge during prediction. A natural choice for measuring this similarity is the categorical classification probability, as it directly reflects the predicted granularity. Specifically, we minimize the probability discrepancy between pixels within an edge $e$ by minimizing the divergence (Kullback & Leibler, 1951) between their granularity probability distributions. To ensure bidirectional discrepancy minimization, we optimize the Jensen-Shannon Divergence (JSD) (Lin, 1991) as follows:

$$l_{\text{consen.}} = \sum_{e \in \mathcal{I}} \sum_{p,q \in e, p \neq q} \text{JSD}(\mathbf{p} \| \mathbf{q}). \tag{4}$$

Here, $p$ and $q$ denote pixels belonging to edge $e$, while $\mathbf{p}, \mathbf{q} \in \mathbb{R}^{|g|}$ represent their respective categorical probability distributions after Softmax normalization. Through derivation in Section A, we obtain a computationally efficient formulation:

$$l_{\text{consen.}} = \sum_{e \in \mathcal{I}} |e| \sum_{p \in e} \mathbf{p} \cdot (\underbrace{\ln(\mathbf{p})}_{\substack{\text{Edge pixel} \\ \text{log-likelihood}}} - \underbrace{\frac{1}{|e|} \sum_{q \in e} \ln(\mathbf{q})}_{\substack{\text{Avg. log-likelihood} \\ \text{of the entire edge}}}). \tag{5}$$

where $\cdot$ represents the dot product. Intuitively, Equation (5) encourages the log-likelihood of each edge pixel $\ln(\mathbf{p})$ to align with the average log-likelihood of the entire edge. When all pixels within the same edge share an identical probability distribution, the consensus loss reaches zero. Since the JSD is always non-negative, this represents the optimal case. The final training objective is a weighted sum of the Junction Confidence-Aware cross-entropy loss and the Edge Consensus Loss, balanced by a trade-off Consensus Loss Weight $\alpha$:

$$\mathcal{L} = l_{\text{entropy}}^{\text{JuncConf}} + \alpha \, l_{\text{consen.}}. \tag{6}$$

## 4 EXPERIMENTS

### 4.1 EXPERIMENTAL SETTINGS

**Dataset and Evaluation Setting.** We evaluate our model on four datasets. The BSDS (Arbelaez et al., 2010b) consists of 200 training, 100 validation, and 200 testing images of natural scenes. The NYUD (Silberman et al., 2012), containing indoor images, includes 381 training, 414 validation, and 654 testing images. The BIPEDv2 (Soria et al., 2023), focused on street view images, has 200 training and 50 testing images, while the Multicue (Mély et al., 2016), featuring outdoor scenes, includes 80 training and 20 testing images. Our evaluation is conducted in a zero-shot setting, meaning that our model is not trained on any images from these datasets. However, for comparison, we also report supervised baselines trained directly on these datasets.

**Implementation Details.** The model is trained with a batch size of 128 for a total of 200,000 iterations with AdamW optimizer (Loshchilov & Hutter, 2017). The learning rate is set to $1 \times 10^{-4}$ with a weight decay of $1 \times 10^{-5}$. Training images are resized to $256 \times 256$ and randomly cropped, while evaluation is performed on full-resolution test images. To avoid aliasing artifacts in edge maps, we apply only minimal data augmentations, including horizontal and vertical flips and 90-degree rotations. To analyze scalability, we test five different model sizes, with detailed configurations provided in Section B. Unless otherwise specified, we use the *base* model size for all experiments. The synthetic dataset is generated using the officially released SAM2 (Ravi et al., 2024). By default, the Consensus Loss weight is set to 0.03 based on our ablation study.

### 4.2 METRICS

**Edge Detection.** Following previous works (Liu et al., 2017; Zhou et al., 2023; 2024; Xie & Tu, 2015), we evaluate edge detection using the F1-score and average precision (AP). The F1-score is

| | BSDS | | | NYUD | | | BIPEDv2 | | | Multicue | | |
|---|---|---|---|---|---|---|---|---|---|---|---|---|
| Method | ODS↑ | OIS↑ | AP↑ | ODS↑ | OIS↑ | AP↑ | ODS↑ | OIS↑ | AP↑ | ODS↑ | OIS↑ | AP↑ |
| *Supervised Trained on Dataset* | | | | | | | | | | | | |
| SE (2014) | 0.746 | 0.767 | 0.803 | 0.695 | 0.708 | 0.679 | - | - | - | - | - | - |
| HED (2015) | 0.788 | 0.808 | 0.840 | 0.720 | 0.734 | 0.734 | 0.829 | 0.847 | 0.869 | 0.851 | 0.864 | - |
| PiDiNet (2021) | 0.789 | 0.803 | - | 0.733 | 0.747 | - | - | - | - | 0.855 | 0.860 | - |
| EDTER (2022) | 0.824 | 0.841 | 0.880 | 0.774 | 0.789 | 0.797 | 0.893 | 0.898 | - | 0.894 | 0.900 | 0.944 |
| UAED (2023)† | 0.829 | 0.847 | 0.892 | - | - | - | - | - | - | 0.895 | 0.902 | 0.949 |
| DiffEdge (2024) | 0.834 | 0.848 | 0.896 | 0.761 | 0.766 | 0.750 | 0.899 | 0.901 | 0.919 | 0.904 | 0.909 | - |
| MuGE (2024)† | 0.831 | 0.847 | 0.886 | - | - | - | - | - | - | 0.898 | 0.900 | 0.950 |
| *Zero-shot* | | | | | | | | | | | | |
| Canny (1986) | 0.600 | 0.640 | 0.580 | 0.438 | 0.438 | 0.336 | 0.664 | 0.665 | 0.560 | 0.689 | 0.690 | 0.598 |
| SAM (2023) | 0.759 | 0.789 | 0.810 | 0.693 | 0.713 | 0.700 | 0.632 | 0.632 | 0.739 | 0.676 | 0.676 | 0.764 |
| UAED (2023)∗ | - | - | - | 0.695 | 0.716 | 0.703 | 0.703 | 0.714 | 0.779 | 0.677 | 0.689 | 0.765 |
| DiffEdge (2024)∗ | 0.725 | 0.735 | 0.723 | 0.689 | 0.711 | 0.702 | 0.734 | 0.748 | 0.807 | 0.714 | 0.718 | 0.770 |
| MuGE (2024)∗ | - | - | - | 0.678 | 0.703 | 0.709 | 0.725 | 0.731 | 0.797 | 0.761 | 0.768 | 0.783 |
| *Zero-shot (Granularly Trained on SGED)* | | | | | | | | | | | | |
| UAED (2023) | 0.709 | 0.756 | 0.698 | 0.662 | 0.670 | 0.652 | 0.764 | 0.770 | 0.809 | 0.805 | 0.808 | 0.822 |
| MuGE (2024) | 0.730 | 0.761 | 0.809 | 0.654 | 0.662 | 0.695 | 0.777 | 0.790 | 0.811 | 0.803 | 0.807 | 0.826 |
| *Zero-shot (Ours)* | | | | | | | | | | | | |
| GEPM (GraLvl=2) | 0.759 | 0.785 | 0.752 | 0.683 | 0.705 | 0.612 | 0.779 | 0.796 | 0.850 | **0.843** | **0.848** | **0.861** |
| GEPM (GraLvl=4) | **0.762** | **0.795** | 0.811 | 0.695 | 0.721 | 0.683 | **0.782** | **0.800** | **0.862** | 0.814 | 0.819 | 0.847 |
| GEPM (GraLvl=6) | **0.762** | 0.794 | **0.821** | **0.701** | **0.723** | **0.715** | 0.773 | 0.790 | 0.854 | 0.795 | 0.799 | 0.822 |
| GEPM (GraLvl=10) | 0.758 | 0.778 | 0.795 | 0.700 | **0.723** | 0.693 | 0.766 | 0.774 | 0.827 | 0.755 | 0.757 | 0.793 |
| GEPM (GraLvl=15) | 0.752 | 0.778 | 0.795 | 0.698 | 0.712 | 0.690 | 0.726 | 0.732 | 0.792 | 0.666 | 0.668 | 0.741 |

† Not trainable on non-granular datasets.

∗ Cross-dataset tested with official weights trained on BSDS by default, and using BIPEDv2 weights for BSDS.

Table 1: Zero-shot edge detection performance on four standard benchmarks, with supervised methods shown for reference. GEPM is evaluated under different Granularity Levels (GraLvl), with its best performance approaching that of some supervised methods. All methods are tested on single-scale image resolution. ↑/↓: higher/lower is better.

computed by binarizing predictions at various thresholds and selecting the best threshold based on precision and recall of edge pixels. This can be determined either per dataset (Optimal Dataset Score, ODS) or per image (Optimal Image Score, OIS). In contrast, AP measures overall performance across different thresholds by averaging precision at varying recall levels. Following (Ye et al., 2024), a predicted pixel is considered a match if it aligns with a ground-truth edge pixel within 1.1% of the image size for NYUD, and 0.75% for other datasets.

**Edge Consistency.** To assess edge consistency, we introduce the Edge Consensus Score and Granularity Variance Score, evaluating how well a model predicts consistent edge-wise granularity across a dataset $\mathcal{D}$. Distinct edges are identified by the graph-based representation in Section 2.2. For each edge $e$ in an image $\mathcal{I}$, we define its predicted label as the most frequently assigned granularity, $e_{\text{Maj.}}$. The Edge Consensus Score is the fraction of pixels agreeing with this majority label, normalized by the total number of edge pixels:

$$\text{Consen.} = \sum_{\mathcal{I} \in \mathcal{D}} \sum_{e \in \mathcal{I}} \sum_{p \in e} \mathbb{I}[p = e_{\text{Maj.}}] / \sum_{\mathcal{I} \in \mathcal{D}} \sum_{e \in \mathcal{I}} |e|, \qquad (7)$$

where $\mathbb{I}$ is the indicator function. The Granularity Variance measures the dispersion of granularity values within each edge, computed as the weighted variance across all edges:

$$\text{GraVar} = \sum_{\mathcal{I} \in \mathcal{D}} \sum_{e \in \mathcal{I}} |e| \text{Var}(e) / \sum_{\mathcal{I} \in \mathcal{D}} \sum_{e \in \mathcal{I}} |e|. \qquad (8)$$

An ideal granular edge prediction model should achieve a high Edge Consensus Score and low Granularity Variance, indicating consistent edge granularity across predictions.

## 4.3 ZERO-SHOT COMPARISON ON BENCHMARKS

We evaluate the zero-shot performance of our model on four edge detection datasets with previous methods' supervised results as a reference. Additionally, we assess the zero-shot generalization ability of three state-of-the-art edge detection models by conducting cross-dataset testing, as well as UAED (Zhou et al., 2023) and MuGE (Zhou et al., 2024)'s zero-shot edge detection performance when they are trained on SGED.

During training, we found that using all 36 granularity values in the SGED edge maps degrades performance. We attribute this to two reasons. First, using all granularity values reduces the annotation frequency of each individual class, which consequently increases the likelihood that the background class dominates the predictions (see Section D for a detailed discussion). Second, having too many granularity values reduces model's tolerance to prediction noise in synthetic annotations. Due to slight variations in edge prominence, similar edges may be labeled with adjacent granularity scores across samples. In this case, a minor prediction error could incur a large penalty, resulting reduced training robustness. To mitigate this issue, we quantize edge granularity into a smaller number of granularity levels by grouping adjacent values. Specifically, for a granularity level (GraLvl) of $x$, all values $v \in [1, 36]$ that share the same floored value $\lfloor \frac{v-1}{36/x} \rfloor$ are grouped together. In total, we test five different granularity levels and present results in Table 1.

Since traditional edge detection is a binary prediction problem, we convert GEPM's categorical predictions into binary predictions for evaluation under standard binary metrics. The conversion process is detailed in Section C.

Empirical results in Table 1 indicate that GEPM outperforms prior zero-shot baselines, especially on densely annotated granular dataset like Multicue. Meanwhile, granularity levels 4 and 6 consistently narrow the gap to fully supervised methods and achieve the best zero-shot edge detection performance across most datasets. These results indicate that i) proper granularity grouping is crucial for zero-shot edge detection generalization, and ii) granular edge priors can transfer robustly for binary edge prediction.

We also observe that baseline methods trained on SGED exhibits better zero-shot performance on BIPED and Multicue compared to when trained on BSDS, but slightly underperforms on NYUD. This asymmetry stems from differences in dataset annotation styles and evaluation compatibility. BSDS and NYUD are both sparsely annotated, focusing primarily on salient object contours, which makes models trained on BSDS naturally generalize better to NYUD. In contrast, SGED shares stronger alignment with densely annotated datasets like BIPED and Multicue due to its fine-grained, multi-level edge annotations. Moreover, evaluating SGED-trained models on NYUD introduces a cross-task mismatch: SGED models are trained for granular prediction, whereas NYUD uses binary metrics. This additional domain gap makes generalization more challenging. Lastly, SGED is synthetically generated, while BSDS is manually labeled, giving BSDS an advantage in perceptual quality, particularly on datasets emphasizing salient edges like NYUD. Despite these factors, SGED-trained models still outperform on BIPED and Multicue, demonstrating the effectiveness of granular supervision and the strong generalization capability of SGED in dense edge prediction scenarios.

## 4.4 GRANULAR EDGE PREDICTION CONSISTENCY

An ideal granular edge prediction model should mimic human annotation behavior by treating edge labeling as a holistic task and ensuring consistent granularity across an edge. We evaluate this consistency using the Edge Consensus Score (Consen.) and Granularity Variance Score (GraVar), as defined in Section 4.2. Our model, GEPM, is compared against two granular prediction baselines, UAED (Zhou et al., 2023) and MuGE (Zhou et al., 2024). Since UAED and MuGE do not directly produce granular predictions, we derive their granular predictions by aggregating their results under different granularity thresholds. Specifically, for

Table 2: Comparison of edge consistency on Consensus Score and Granularity Variance.

| Method | | UAED | MuGE | GEPM |
|---|---|---|---|---|
| **BSDS** | Consen.↑ | 0.655 | 0.728 | **0.910** |
| | GraVar↓ | 0.622 | 0.612 | **0.121** |
| **NYUD** | Consen.↑ | 0.655 | 0.706 | **0.881** |
| | GraVar↓ | 0.600 | 0.583 | **0.148** |
| **BIPEDv2** | Consen.↑ | 0.626 | 0.703 | **0.893** |
| | GraVar↓ | 0.675 | 0.555 | **0.173** |
| **Multicue** | Consen.↑ | 0.776 | 0.718 | **0.911** |
| | GraVar↓ | 0.592 | 0.538 | **0.131** |

UAED, we perform multiple stochastic samplings on edge pixels, while for MuGE, we evenly sample conditioning granularity thresholds. The assigned granularity for an edge pixel is determined by the number of times it is recognized across these settings. To ensure a fair comparison, we set their granularity settings to 6, matching GEPM's configuration.

As shown in Table 2, GEPM achieves substantially higher consistency scores compared to both baselines. This improvement stems from explicitly modeling edge granularity during training and enforcing structured predictions through the proposed edge consensus loss. These findings highlight GEPM's effectiveness in producing more coherent, human-aligned predictions. Furthermore, qualitative results in Figure 4 illustrate that GEPM better preserves consistent granularity across entire edges, whereas baselines often assign varying granularities to different pixels on the same edge.

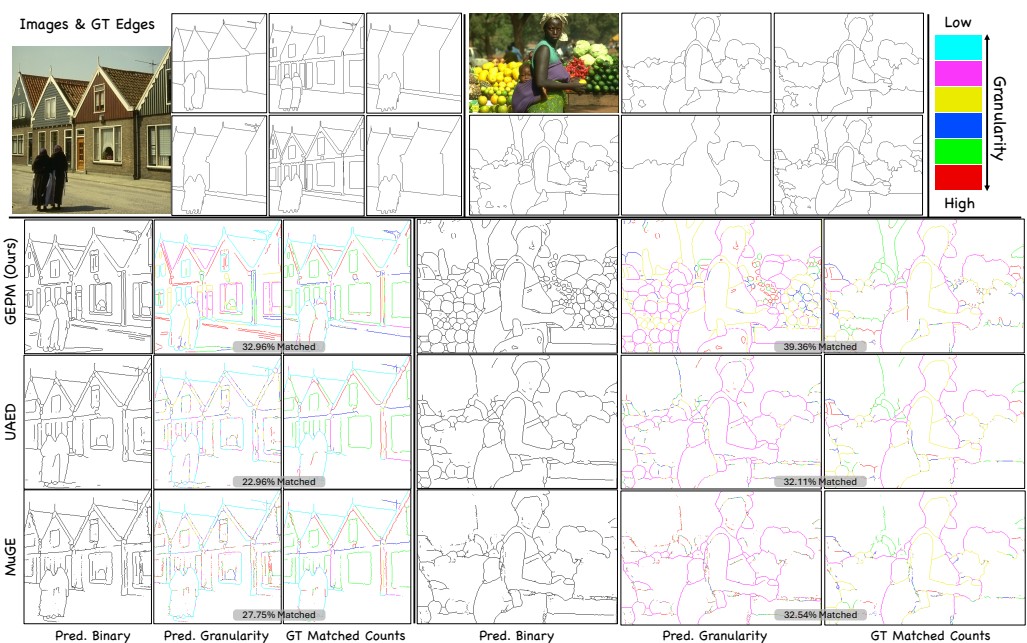

Figure 4: Granular edge detection on BSDS. Ground-truth labels from different annotators are shown. We visualize the binarized prediction (left), granular prediction (middle), and the occurrence of binarized predictions matching human annotations (right) with matching percentage in the middle. Ideally, edge granularity should correspond to its match count with human annotations. Compared to baselines, GEPM achieves 1) more consistent edge granularity, 2) higher edge recall, and 3) better alignment with ground truth. Better viewed digitally with zoom.

### 4.5 ABLATION STUDIES

We conduct ablation studies using the BSDS dataset. Since our model is evaluated in a zero-shot setting, similar conclusions can be drawn for other datasets.

**On Consensus Loss** We analyze the impact of the Consensus Loss Weight $\alpha$ on granular edge prediction performance, with quantitative results presented in Figure 5(a) and (b). We evaluate five values of $\alpha$ ranging from 0 to 0.3, where $\alpha = 0$ corresponds to optimization solely with cross-entropy loss. Increasing $\alpha$ leads to a higher Edge Consensus Score and lower Granularity Variance, indicating improved edge-wise granularity consistency. However, this comes at the cost of reduced performance on edge detection, as reflected by decreases in OIS, ODS, and AP scores. This tendency is also observed in the visualization in Figure 6. As consensus loss increases, it dominates the loss and degrades model's ability to detect edges. Based on these findings, we recommend $\alpha = 0.03$ or $\alpha = 0.1$, as these values enhance edge consistency without significantly compromising edge detection performance.

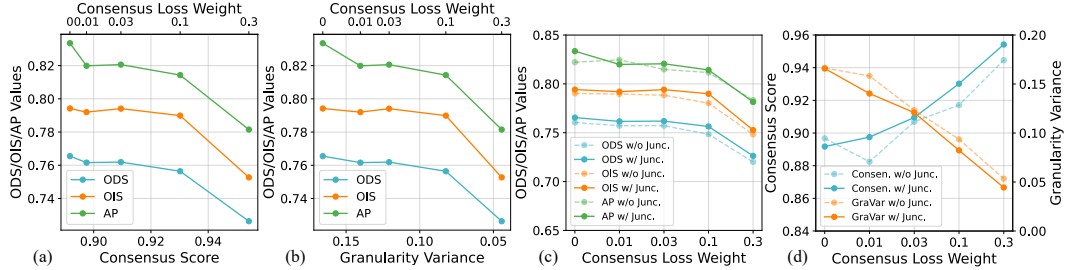

Figure 5: (a, b) Performance trade-off between edge detection (ODS/OIS/AP) and edge consistency (Consensus Score, Granularity Variance) at different consensus loss weights $\alpha$. (c, d) Junction Confidence-Aware training (solid line) improves both edge detection and edge consistency across various consensus loss weights.

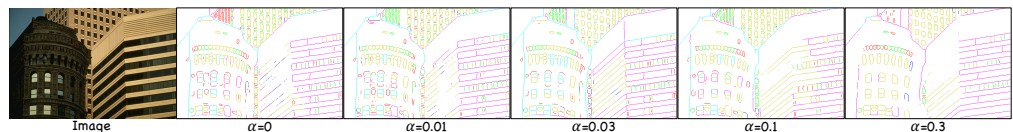

Figure 6: Visualization of GEPM trained with different consensus loss weights. Increasing the consensus loss weight improves edge consistency but reduces model's edge detection capability.

**On Junction Confidence-Aware Training** We observe that blocking gradients from regions with an unusually excessive junction points benefits GEPM training from Figure 5(c) and (d). These regions often correspond to ambiguous edges where our synthetic annotation method is prone to errors. Experimental results confirm that junction confidence-aware training improves both edge detection and edge consistency performance across different consensus loss weights, demonstrating its effectiveness in mitigating annotation noise and enhancing model robustness.

### 4.6 Granularity-Aware Edge Evaluation

The evaluations conducted in the previous sections follow the standard binary edge detection protocol. However, in the context of Granular Edge Prediction, it is equally important to ensure that the predicted granularity aligns with human perception. To address this, we introduce Granularity-Aware Edge Evaluation, a metric that measures the discrepancy between a pixel's predicted granularity and its occurrence in ground-truth annotations from multiple annotators. For a perfectly matched granularity, this value should be zero, with lower values indicating better alignment. The colorized edge map in Figure 4 visualizes the granular predictions alongside their matching occurrence with human annotations.

We conduct experiments on BSDS using different consensus loss weights and model sizes, all trained with a granularity level of 6. For comparison, we use the granular edge prediction models directly trained on BSDS. The results, presented in Figure 7, categorize pixels into perfectly matched, matched with a small difference (granularity difference 1-2), and badly matched groups (granularity difference 3+

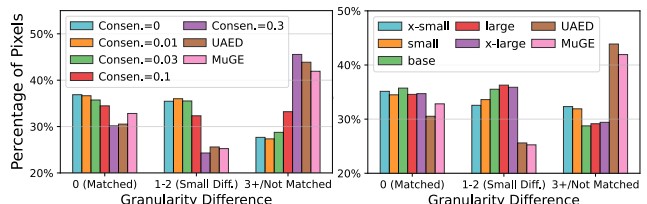

Figure 7: Granularity matching distribution for GEPM and baselines on BSDS. GEPM outperforms baselines, even when baselines are directly optimized on BSDS.

or not matched). The influence of consensus loss weight follows a trade-off similar to earlier evaluations between edge detection and edge consistency. With a moderate consensus loss, the proportion of correctly matched granularities remains high. However, excessively large weights increase the proportion of badly matched granularities. This observation further underscores the importance of adopting an appropriate consensus loss during training.

Regarding model size, the results indicate that while the percentage of perfectly matched predictions remains similar across different model sizes, larger models show improvements in cases with small granularity differences. This improvement diminishes when increasing model size from *large* to *x-large*. We hypothesize two possible reasons for this observation: 1) The SGED dataset size is limited, and the *large* model already captures most of its distribution, so further increasing model capacity does not yield additional benefits. 2) The synthetic labels in SGED exhibit inherent discrepancies compared to real human annotations, and increasing model size alone cannot bridge this gap. These two factors are not mutually exclusive and may both contribute to the observed performance saturation. Addressing these limitations requires improvements in the dataset, which may exceed the capabilities of the current synthetic generation pipeline and necessitate human annotation, a direction we leave for future work.

## 5 CONCLUSION

In this work, we investigate the problem of granular edge prediction. Recognizing the scarcity of suitable datasets for training granular edge prediction models, we construct a large-scale synthetic granular edge dataset by leveraging existing segmentation models. By representing edges as a graph, we address the issue of label inconsistency in the coarsely generated dataset. We propose a granular edge prediction model that simultaneously estimates edges and their corresponding granularity by formulating edge detection as a categorical classification problem. To ensure granular consistency, we introduce the Edge Consensus Loss and Junction Confidence-Aware Training, improving alignment between predictions and human perception. Extensive experiments analyze the effects of training granularity, consensus loss weighting, and model size. Zero-shot evaluations on both standard binary edge detection benchmarks and granularity-aware evaluation metrics demonstrate the effectiveness of the proposed approach.

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

# Appendix for Generalizable And Consistent Granular Edge Prediction

## A    DERIVATION FOR EDGE CONSENSUS LOSS

We provide the derivation of the Jensen-Shannon Divergence (JSD) between pixels belonging to the same edge. As a reminder, let $e$ represent an edge (a set of pixels) obtained using the graph representation described in Section 2.2. Let $p$ and $q$ be pixels within edge $e$, and let $\mathbf{p}, \mathbf{q} \in \mathbb{R}^{|g|}$ denote their corresponding granular categorical probability distributions after Softmax normalization, where $|g|$ is the total number of granularity levels including the background.

$$\sum_{p,q\in e, p\neq q} \text{JSD}(\mathbf{p}||\mathbf{q}) \stackrel{\text{①}}{=} \sum_{p,q\in e, p\neq q} \text{JSD}(\mathbf{p}||\mathbf{q}) + \sum_{p\in e} \text{JSD}(\mathbf{p}||\mathbf{p}) \tag{9}$$

$$= \sum_{p,q\in e} \text{JSD}(\mathbf{p}||\mathbf{q}) \tag{10}$$

$$\stackrel{\text{②}}{=} \sum_{p,q\in e} \frac{1}{2}\text{KLD}(\mathbf{p}||\mathbf{q}) + \frac{1}{2}\text{KLD}(\mathbf{q}||\mathbf{p}) \tag{11}$$

$$= \sum_{p,q\in e} \text{KLD}(\mathbf{p}||\mathbf{q}) \tag{12}$$

$$= \sum_{p,q\in e} \mathbf{p} \cdot \ln\frac{\mathbf{p}}{\mathbf{q}} \tag{13}$$

$$= \sum_{p\in e} \mathbf{p} \cdot \sum_{q\in e} (\ln(\mathbf{p}) - \ln(\mathbf{q})) \tag{14}$$

$$= \sum_{p\in e} \mathbf{p} \cdot \left( |e|\ln(\mathbf{p}) - \sum_{q\in e} \ln(\mathbf{q}) \right) \tag{15}$$

$$= |e| \sum_{p\in e} \mathbf{p} \cdot \left( \ln(\mathbf{p}) - \frac{1}{|e|} \sum_{q\in e} \ln(\mathbf{q}) \right) \tag{16}$$

For clarity, we use $\sum_{p,q\in e}$ as shorthand for $\sum_{p\in e} \sum_{q\in e}$. Equation ① holds because the divergence between two identical distributions is zero, *i.e.*, $\text{JSD}(\mathbf{p}||\mathbf{p}) = 0$. Equation ② reformulates the bidirectional JSD in terms of the Kullback-Leibler Divergence (KLD) based on its definition.

## B    MODEL DETAILS

We evaluate five different model size configurations, namely *x-small*, *small*, *base*, *large*, and *x-large*, with detailed specifications provided in Table 3. For brevity, we present only the down-scaling block structures, as the corresponding up-scaling structures are symmetric. The model architecture follows the naming conventions used in the Python package `diffusers`[1].

It is worth noting that we only adopt the U-Net backbone component from `diffusers.UNet2DModel` for its simplicity, modularity, and ease of future reproducibility. GEPM is trained entirely from scratch. No pretrained weights from diffusion models are used during training. Our usage of the `diffusers` codebase is purely for constructing the architecture and does not involve any generative or diffusion-based training objective. The full architecture can be exactly reconstructed by instantiating `diffusers.UNet2DModel` using the configuration we provide.

We also examine the impact of GEPM's model size on binary edge detection and consistency performance with results presented in Table 4. At first glance, increasing model size does not yield

---

[1] `https://huggingface.co/docs/diffusers/en/index`

Table 3: The five model configuration of different sizes use in our experiments. The block is built with and named under the manner of Python package `diffusers`

| Model | Structure | Layer Channels | Layers per Block | Total Number of Params. |
|---|---|---|---|---|
| x-small | DownBlock2D | 32 | 2 | 93,247,687 |
| | DownBlock2D | 64 | | |
| | DownBlock2D | 128 | | |
| | DownBlock2D | 256 | | |
| | DownBlock2D | 512 | | |
| | DownBlock2D | 512 | | |
| small | DownBlock2D | 32 | 2 | 182,500,039 |
| | DownBlock2D | 64 | | |
| | DownBlock2D | 128 | | |
| | DownBlock2D | 256 | | |
| | DownBlock2D | 512 | | |
| | DownBlock2D | 896 | | |
| base | DownBlock2D | 64 | 2 | 282,262,919 |
| | DownBlock2D | 128 | | |
| | DownBlock2D | 256 | | |
| | DownBlock2D | 512 | | |
| | DownBlock2D | 512 | | |
| | AttnDownBlock2D | 1024 | | |
| large | DownBlock2D | 64 | 3 | 369,951,559 |
| | DownBlock2D | 128 | | |
| | DownBlock2D | 256 | | |
| | DownBlock2D | 512 | | |
| | AttnDownBlock2D | 512 | | |
| | AttnDownBlock2D | 1024 | | |
| x-large | DownBlock2D | 64 | 4 | 467,105,031 |
| | DownBlock2D | 128 | | |
| | DownBlock2D | 256 | | |
| | DownBlock2D | 512 | | |
| | AttnDownBlock2D | 512 | | |
| | AttnDownBlock2D | 1024 | | |

significant improvements in edge detection performance, while edge consistency shows noticeable enhancement. While this may suggest that model size primarily benefits edge consistency with minimal effect on edge detection, it is important to note that edge detection is evaluated under the traditional binary protocol. According to the finding in Section 4.6, we notice that larger models still enhance edge detection by improving their capability for granularity-aware prediction, which is not fully captured in binary evaluation metrics.

Table 4: Edge detection and consistency performance of GEPM across different model sizes.

| Model Size | ODS↑ | OIS↑ | AP↑ | Consen.↑ | GraVar↓ |
|---|---|---|---|---|---|
| x-small | 0.756 | 0.786 | 0.824 | 0.887 | 0.159 |
| small | 0.757 | 0.785 | 0.825 | 0.905 | 0.129 |
| base | 0.762 | 0.794 | 0.821 | 0.910 | 0.121 |
| large | 0.756 | 0.788 | 0.826 | 0.914 | 0.111 |
| x-large | 0.762 | 0.797 | 0.826 | 0.915 | 0.110 |

## C    CONVERTING GRANULAR PREDICTION TO BINARY PREDICTION

In **Zero-shot Comparison on Benchmarks**, we evaluate our model on four standard edge detection datasets, where the benchmarks require binary edge predictions with probability values ranging from 0 to 1. However, our Granular Edge Prediction Model (GEPM) produces categorical predictions, making direct evaluation incompatible with these benchmarks. To address this, we implement a

straightforward conversion method to transform categorical predictions into binary edge maps, as outlined in Figure 8.

Our conversion strategy accounts for two key factors: (1) Edge granularity, where edges with lower granularity (*i.e.*, frequently annotated edges) correspond to higher class indices in the predicted probability distribution `pred_prob`, and should have binarized probabilities closer to 1. Conversely, edges with higher granularity (*i.e.*, less frequently annotated edges) should have lower binarized probabilities. (2) Granularity adjustment, where we adjust a pixel's probability based on its second-most likely granularity class. If the second probable granularity is lower (indicating a more salient edge), we increase its binarized probability; otherwise, we decrease it.

To implement this, we define a minimum and maximum binarized probability for each granularity level. For any two consecutive granularity levels, the minimum probability of a lower granularity edge is set as the maximum probability of the higher granularity edge. Furthermore, within each granularity level, we refine the binarized probability using the secondary granularity class (as described in Figure 8, line 24) and normalize it within the corresponding granularity range. This approach ensures a smooth transition between granularity levels on the converted binary edge map.

We also tested two other straightforward granular-to-binary edge conversion methods. 1) Linear Class Index mapping and 2) Average Probability mapping. The former one uses the highest granular probability and linearly maps it to a value between 0 to 1 according to the granularity levels (*e.g.*, in GraLvl=6, we have mapping categorical $\rightarrow$ binary probability with $0 \rightarrow 0$, $1 \rightarrow 1/6$, ..., $6 \rightarrow 1$). The latter one use the weighted average of granularity where weights are the categorical probabilities of the granular prediction. Figure 9 and Table 5 demonstrates the visual and quantitative comparison between our granular-to-binary conversion method and two straightforward approaches. We found that the linear class index mapping yielded suboptimal performance due to the loss of relative edge probability within each granularity level. While the average probability mapping quantitatively performs comparably to our strategy, we observe that this strategy introduces a "blurring" effect, whereas our method produces sharper edge maps.

| Metrics | Ours | Linear | Avg Prob. |
|---------|------|--------|-----------|
| ODS | **0.762** | 0.752 | 0.761 |
| OIS | **0.794** | 0.787 | **0.794** |
| AP | 0.821 | 0.806 | **0.824** |

Table 5: Quantitative comparison of three different granular-to-binary edge conversion methods.

## D  LIMITATIONS AND FUTURE WORK

While our method demonstrates strong zero-shot performance, several limitations remain. First, we adopt a UNet as backbone for its simplicity and have not explored different backbone architectures, some of which may yield better results. However, we want to highlight that our method is backbone-agnostic, which should be seen as an advantage as it gets rid of heavy empirical model design in edge detection. Additionally, advanced training strategies, such as diffusion-based training Ho et al. (2020), could potentially enhance edge sharpness. Although our approach leads in zero-shot testing, it does not surpass supervised training when the test dataset distribution is known. In such cases, supervised training remains superior. A promising direction for future work is developing efficient fine-tuning strategies to adapt our model to specific datasets when zero-shot testing is not required. We believe our model can serve as a strong foundational edge predictor and achieve state-of-the-art performance after proper fine-tuning.

Also, we observe that GEPM may experience dataset-dependent performance degradation under certain extreme granularity levels. In the Table 6, we evaluate GEPM using two additional granularity levels. The GraLvl=1 case reduces the granular prediction to a binary prediction problem and the GraLvl=36 case retains all granularity values from the SGED dataset. Specifically, when trained with a less granularity levels (GraLvl=1), GEPM shows reduced performance on BSDS and NYUD. Conversely, when trained with more granularity levels (GraLvl=36), the performance declines on BIPEDv2 and Multicue.

```python
def multiclass_to_binaryclass(pred_prob):
    # pred_prob: (num_classes, H, W) num class inlcude a background
    eps = 5e-3 # half of 1e-2
    num_classes = pred_prob.shape[0] - 1
    binarized_edge = np.zeros_like(pred_prob[0])

    top1_prob_class = np.argmax(pred_prob, axis=0)
    top2_prob_class = np.argsort(pred_prob, axis=0)[-2]
    top1_prob = pred_prob[top1_prob_class, np.arange(pred_prob.shape[1])
        [:, None], np.arange(pred_prob.shape[2])]
    top2_prob = pred_prob[top2_prob_class, np.arange(pred_prob.shape[1])
        [:, None], np.arange(pred_prob.shape[2])]

    prob_diff = np.where(top1_prob_class - top2_prob_class > 0, -1, 1) #
        if top1_prob_class > top2_prob_class, (e.g., 0.3, 0.4, 0.1, ...),
         we should adjust the edge to smaller value, vice versa

    for class_id in range(1, num_classes+1):
        this_min_val = (class_id - 1) / num_classes
        this_max_val = class_id / num_classes
        budget_each_class = 1 / num_classes

        this_class_loc = top1_prob_class == class_id

        if this_class_loc.sum() == 0:
            continue
        this_class_prob = pred_prob[class_id]
        this_class_prob_adjusted = this_class_prob + prob_diff * (
            this_class_prob - top2_prob) # adjust the edge to smaller
            value if top1_prob_class > top2_prob_class, vice versa
        this_class_min_val = this_class_prob_adjusted[this_class_loc].min
            ()
        this_class_max_val = this_class_prob_adjusted[this_class_loc].max
            ()
        normalized_class_prob = (this_class_prob_adjusted -
            this_class_min_val) / (this_class_max_val -
            this_class_min_val + eps)
        normalized_class_prob = normalized_class_prob * budget_each_class
             + this_min_val
        normalized_class_prob = np.clip(normalized_class_prob,
            this_min_val + eps, this_max_val - eps)
        binarized_edge = np.where(this_class_loc, normalized_class_prob,
            binarized_edge)

    return binarized_edge
```

Figure 8: The `Python` code to convert a categorical granular edge prediction to a binary edge prediction.

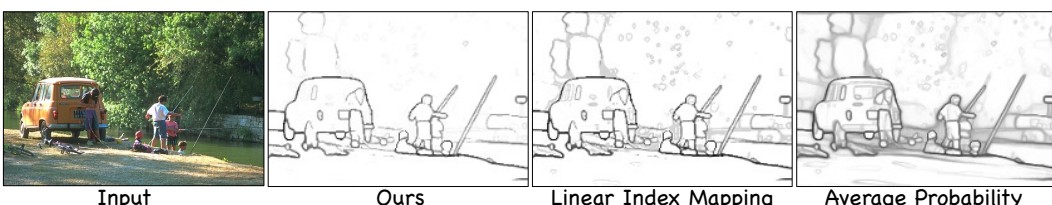

| Input | Ours | Linear Index Mapping | Average Probability |

Figure 9: Visual comparison of three different granular-to-binary edge conversion methods. While the average probability mapping quantitatively performs comparably to our strategy, we observe that this strategy introduces a "blurring" effect, whereas our method produces sharper edge maps.

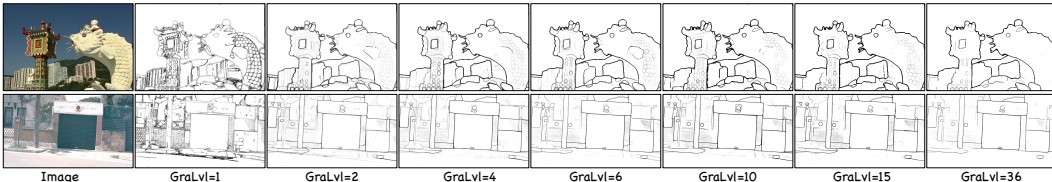

Figure 10: Predictions of GEPM trained with different granularity levels. Models trained with fewer granularity levels tend to detect more edges, which reduces precision on sparsely annotated datasets such as BSDS and NYUD. In contrast, models trained with more granularity levels are prone to being dominated by the background class, leading to lower edge recall. This results in degraded performance on densely annotated datasets like BIPEDv2 and Multicue.

| | BSDS | | | NYUD | | | BIPEDv2 | | | Multicue | | |
|---|---|---|---|---|---|---|---|---|---|---|---|---|
| Method | ODS↑ | OIS↑ | AP↑ | ODS↑ | OIS↑ | AP↑ | ODS↑ | OIS↑ | AP↑ | ODS↑ | OIS↑ | AP↑ |
| GEPM (GraLvl=1) | 0.662 | 0.678 | 0.516 | 0.611 | 0.619 | 0.487 | 0.774 | 0.783 | 0.738 | 0.838 | 0.846 | 0.831 |
| GEPM (GraLvl=36) | 0.740 | 0.762 | 0.792 | 0.680 | 0.695 | 0.714 | 0.649 | 0.651 | 0.748 | 0.548 | 0.549 | 0.676 |
| GEPM (Best) | 0.762 | 0.795 | 0.821 | 0.701 | 0.723 | 0.715 | 0.782 | 0.800 | 0.862 | 0.843 | 0.848 | 0.861 |

Table 6: GEPM's zero-shot edge detection performance may have dataset-dependent degradation when using GraLvl=1 or GraLvl=36. This is a shortcoming primarily due to the mismatch between the prediction and annotation edge density. We suggest to train GEPM with intermediate granularity levels as they are more robust to datasets of different annotation density.

We find that this degradation is primarily due to a mismatch between the prediction and annotation edge density. At GraLvl=36, the model must distinguish between many edge classes, reducing the frequency of each individual class. This increases the likelihood that the background class dominates the predictions, resulting in lower edge recall (as evident in Figure 10). As BIPEDv2 and Multicue have dense annotations, this low recall leads to significant performance drops. On the other hand, at GraLvl=1, all edge pixels are grouped into a single class. This increases the chance of correctly predicting edge presence, improving recall but reducing precision due to increased false positives. For sparsely annotated datasets like BSDS and NYUD, this precision drop has a greater impact, ultimately degrading performance. However, we want to highlight that that GEPM achieves balanced and robust performance across datasets at intermediate granularity levels.

To validate GEPM's potential for downstream tasks, we conduct two experiments. One is the feature probing to asses the feature quality of GEPM by fine-tuning it on the binary edge prediction task. The second is applying the predicted granular on a downstream depth estimation task.

We conduct a feature probing experiment using a two-layer MLP on the "base" GEPM (GraLvl=6) model to assess its suitability for downstream tasks. Specifically, we replace the final layer with a binary prediction head (the two-layer MLP) and freeze all other pretrained layers. This setup is both parameter-efficient and evaluates the quality of learned features. The total number of trainable parameters is 20K (0.007% of the full model). The fine-tuned model improves ODS/OIS/AP by 4.35%/2.72%/2.34%, narrowing the gap to the SoTA (supervisedly trained on BSDS) by 60.4%/50.4%/31.2%, respectively. We believe further gains are possible with more sophisticated training strategies.

To validate the utility of GEPM's granular outputs, we conducted a depth estimation experiment based on the publicly available implementation of BTS (Lee et al., 2019).[2] This method can be trained on a single dataset, making it a suitable choice to isolate the impact of adding granular edge priors without interference from mixed training sources.

In our experiment, we train the BTS model on the NYUD depth estimation task. We use the GEPM model with GraLvl=6 to produce zero-shot granular edge predictions for NYUD images, as this configuration yields the best edge detection performance on NYUD as shown in Table 1. To inject the edge information into the BTS network, we concatenate the predicted granular edge map

---

[2]https://github.com/cleinc/bts

to each of the last four DenseNet feature stages of BTS after downscaling it using a lightweight 5-layer convolutional projection module, each with just 8 feature channels. This setup allows granular edge information to be integrated across multiple scales without significantly increasing model complexity.

As shown in Table 7, injecting GEPM's granular predictions leads to modest but consistent improvements across most evaluation metrics, confirming the value of granularity-aware edge priors for downstream dense prediction tasks. A qualitative comparison in Figure 11 further demonstrates this benefit. By incorporating granular edges, the predicted depth map preserves sharper boundaries and clearer geometric structures, which are typically smoothed or lost in the baseline prediction.

| Method | $\delta_1 \uparrow$ | $\delta_2 \uparrow$ | $\delta_3 \uparrow$ | AbsRel $\downarrow$ | SqRel $\downarrow$ | RMSE $\downarrow$ | RMSElog $\downarrow$ | SILog $\downarrow$ | log10 $\downarrow$ |
|---|---|---|---|---|---|---|---|---|---|
| BTS (Baseline) | 0.878 | 0.979 | 0.995 | 0.113 | 0.069 | 0.400 | 0.144 | 11.772 | 0.048 |
| + Granular Edge | 0.879 | 0.979 | 0.996 | 0.111 | 0.065 | 0.398 | 0.143 | 11.600 | 0.048 |

Table 7: Depth estimation performance on NYUD dataset with and without GEPM-generated granular edges. When training, we use the same hyper-parameters and network initialization.

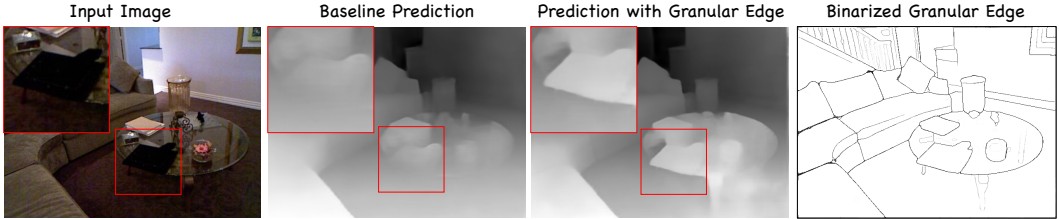

Figure 11: Qualitative comparison on NYUD depth estimation. The injected granular edges help preserve object contours and improve depth boundary sharpness.

## E  CONFIGURATION FOR SEGMENT ANYTHING MODEL TO GENERATE SYNTHETIC EDGES

We experimented with various hyperparameter settings in the Segment Anything Model (SAM) and identified parameters that decently impact the generation of synthetic edges. Based on our findings, we selected the following values:

- `points_per_side` = [16, 32, 64]
- `stability_score_thresh` = [0.85, 0.88, 0.90, 0.92, 0.94, 0.96]
- `crop_n_layer` = [1, 2]

We settled on these values for the following reasons. For `points_per_side`, we adopted the default values recommended by SAM, as increasing this parameter beyond 64 leads to a polynomial increase in computational cost without noticeably improving edge quality. For `crop_n_layer`, we observed that increasing it beyond 2 yields negligible changes in the detected edge pixels.

The setting of `stability_score_thresh` required more careful tuning. We empirically identified the effective range to be [0.85, 0.96], where lowering the threshold below 0.85 fails to yield additional edges, and increasing it above 0.96 often results in empty outputs. Within this range, we uniformly sampled six values at intervals of 0.02. Smaller intervals (*e.g.*, 0.01) produced highly similar edge maps, reducing dataset diversity while greatly increasing collection cost, whereas larger intervals caused abrupt shifts in edge content, reducing the smoothness of granularity progression.

As a result, our 36 granularity levels arise naturally as the product of the above settings: 3 (`points_per_side`) × 2 (`crop_n_layer`) × 6 (`stability_score_thresh`) = 36. These combinations collectively define our granularity levels, providing a stable and smoothly varying granularity spectrum.

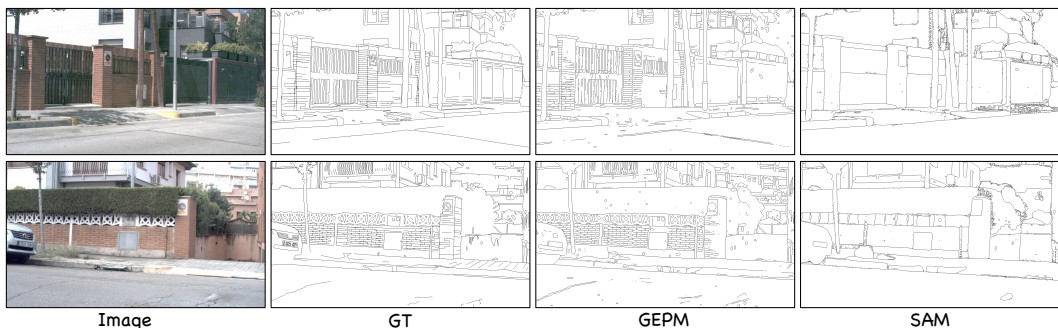

Figure 12: The binary prediction comparison between GEPM and SAM. While GEPM is trained on synthetic data generated by SAM, it generalizes and performs better than SAM on dataset with detailed annotations.

Figure 13 illustrates how different values of points_per_side and stability_score_thresh impact the resulting synthetic edges.

## F    DISCUSSION OF SAM'S ZERO-SHOT PERFORMANCE ON EDGE DETECTION AND COMPARISON WITH GEPM

Deep learning models are inherently data-driven, meaning their performance is largely determined by the quality of the training data. In this section, we aim to answer a key question: is the GEPM model simply replicating the behavior of a "naive SAM zero-shot edge detector," or does it generalize beyond the data distribution created by SAM?

Here, we define a naive SAM zero-shot edge detector as a multi-step edge detection approach that replicates the same procedure used to generate the SGED dataset. While such a detector is impractical for real-world applications – requiring multiple runs of SAM under different configurations followed by extensive post-processing, making inference time prohibitively slow (minutes per image) – it remains valuable to understand its theoretical performance and how it compares to GEPM, which is trained on its generated data.

We compare the naive SAM detector and GEPM using ODS, OIS, and AP scores under the standard binary evaluation protocol, as well as Granularity-Aware Edge Evaluation in Section 4.6. Edge consistency metrics are excluded from this comparison, as the post-processing in SGED ensures perfect edge consensus with zero granularity variance, making such an evaluation meaningless. Both models are evaluated at Granularity Level 6 (GraLvl=6).

The results, presented in Table 8 and Table 9, demonstrate that GEPM consistently outperforms the naive SAM detector across all aspects. In BSDS and NYUD, the binary evaluation gap between the two models is relatively small, whereas the gap is significantly larger in BIPEDv2 and Multi-cue. More importantly, in granularity-aware evaluation, GEPM achieves a much higher number of matched pixels, confirming that GEPM is not merely a direct "student" of SAM or its derived SGED dataset but instead generalizes as a more robust edge detector.

However, this raises a question about how this performance improvement comes. After a careful inspection, we attribute its advantage to several key factors. 1) First, SAM primarily detects object boundaries, meaning if an edge does not correspond to a distinct object – such as fabric patterns – SAM often fails to capture it. In contrast, GEPM generalizes better through training, aided by random cropping and augmentation, which shifts the model's focus from object contours to edge structures. 2) Second, SAM's performance is highly dependent on its hyperparameters and the density of objects in an image. For instance, in images with many objects, effective edge detection requires a higher number of sampling points (the hyperparameter points_per_side mentioned in Section E). This issue is particularly evident in outdoor scene datasets like BIPEDv2 and Multi-cue, where SAM struggles to recall all edge candidates, leading to a significant performance gap compared to GEPM, as evident in Figure 12. Conversely, datasets like BSDS and NYUD, which

contain fewer object-related complexities, show a smaller gap. Since GEPM is trained in a pixel-wise manner, it remains unaffected by excessive objects or edge density in test images. 3) Third, SAM produces predictions on a per-instance basis, followed by strict binarization after granularity quantization. In contrast, GEPM outputs categorical probability distributions, offering a statistical summary of granularity likelihood over the entire SGED dataset. This probabilistic modeling enables GEPM to better reflect granularity uncertainty, leading to improved evaluation performance across varying thresholds.

Based on these observations, we conclude that GEPM is not merely a distribution student of SAM but has generalized into a more robust and versatile edge detector. By learning beyond the object-centric nature of SAM and overcoming its hyperparameter sensitivity, GEPM demonstrates superior performance in both binary and granularity-aware evaluations.

Table 8: Binarized edge detection performance comparison between naive SAM edge detector and GEPM

| Method | | Naive SAM | GEPM |
|---|---|---|---|
| **BSDS** | ODS↑ | 0.7593 | 0.7619 |
| | OIS↑ | 0.7893 | 0.7941 |
| | AP↑ | 0.8120 | 0.8206 |
| **NYUD** | ODS↑ | 0.6932 | 0.7008 |
| | OIS↑ | 0.7129 | 0.7230 |
| | AP↑ | 0.7007 | 0.7149 |
| **BIPEDv2** | ODS↑ | 0.6317 | 0.7733 |
| | OIS↑ | 0.6323 | 0.7900 |
| | AP↑ | 0.7393 | 0.8537 |
| **Multicue** | ODS↑ | 0.6756 | 0.7952 |
| | OIS↑ | 0.6757 | 0.7985 |
| | AP↑ | 0.7638 | 0.8216 |

Table 9: Granularity-aware edge evaluation comparison between naive SAM edge detector and GEPM on BSDS

| | Granularity Difference | | |
|---|---|---|---|
| Method | 0 (Matched) | 1-2 (Small Diff.) | 3+/Not Matched |
| Naive SAM | 0.2884 | 0.3453 | 0.3662 |
| GEPM | 0.3573 | 0.3552 | 0.2875 |

# G  EXAMPLES OF SYNTHETIC GRANULAR EDGE DATASET

We show random samples of generated Synthetic Granular Edge Dataset in Figures 14 to 16 with source images from LAION and their synthetic granular edge side-by-side. We center crop the image to make the aspect ratio to be one for better visualization, the actual source image and corresponding edge in the dataset has preserve their original aspect ratio.

# H  ACKNOWLEDGMENT

The authors would like to acknowledge the use of ChatGPT for assistance in improving the clarity and readability of the manuscript. ChatGPT was employed to refine language expression and correct grammatical errors. All scientific content, experimental design, analysis, and conclusions remain the sole responsibility of the authors.

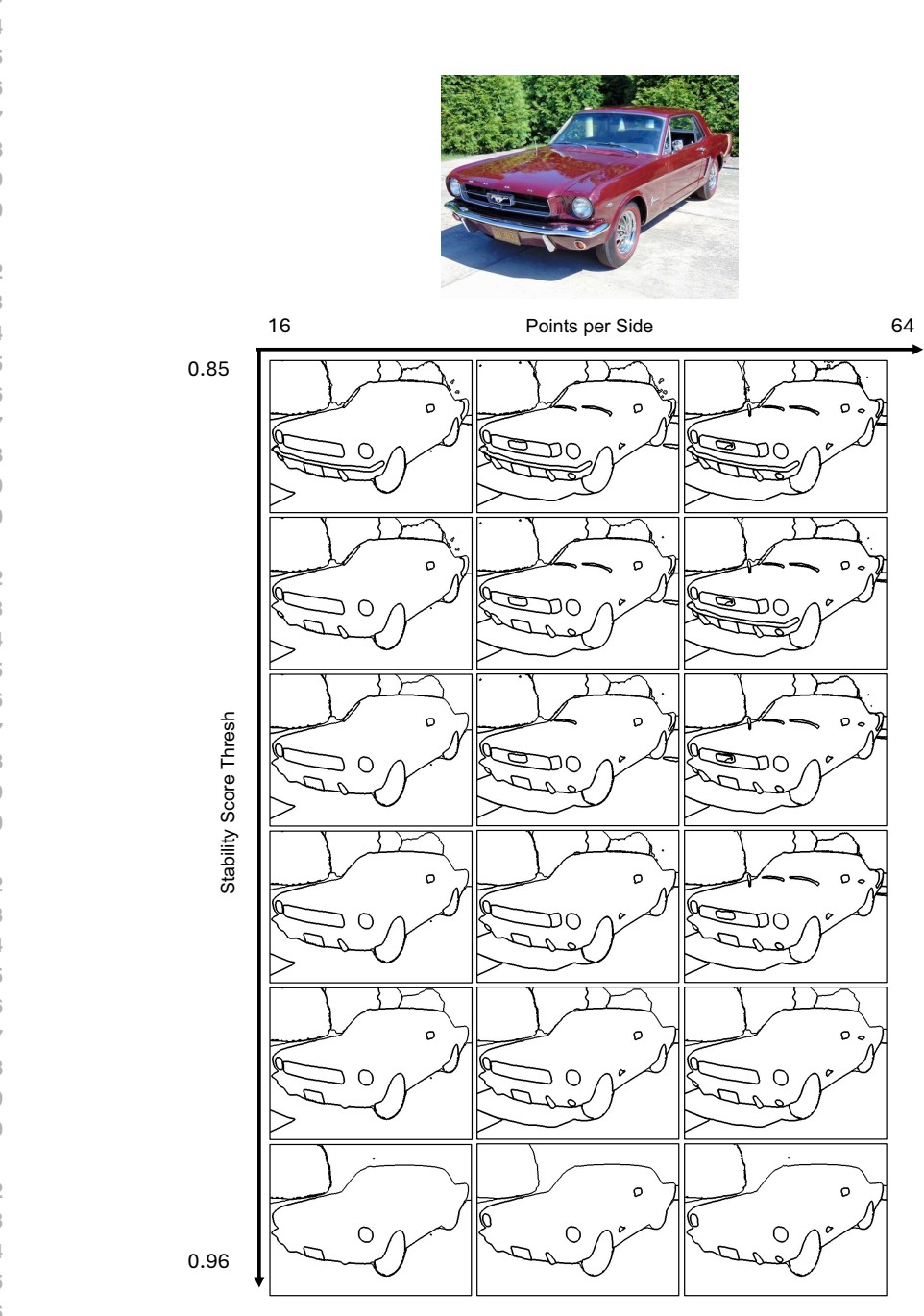

Figure 13: Effect of SAM hyperparameters on synthetic edges generation. We illustrate the impact of `points_per_side` and `stability_score_thresh`, the two most influential parameters.

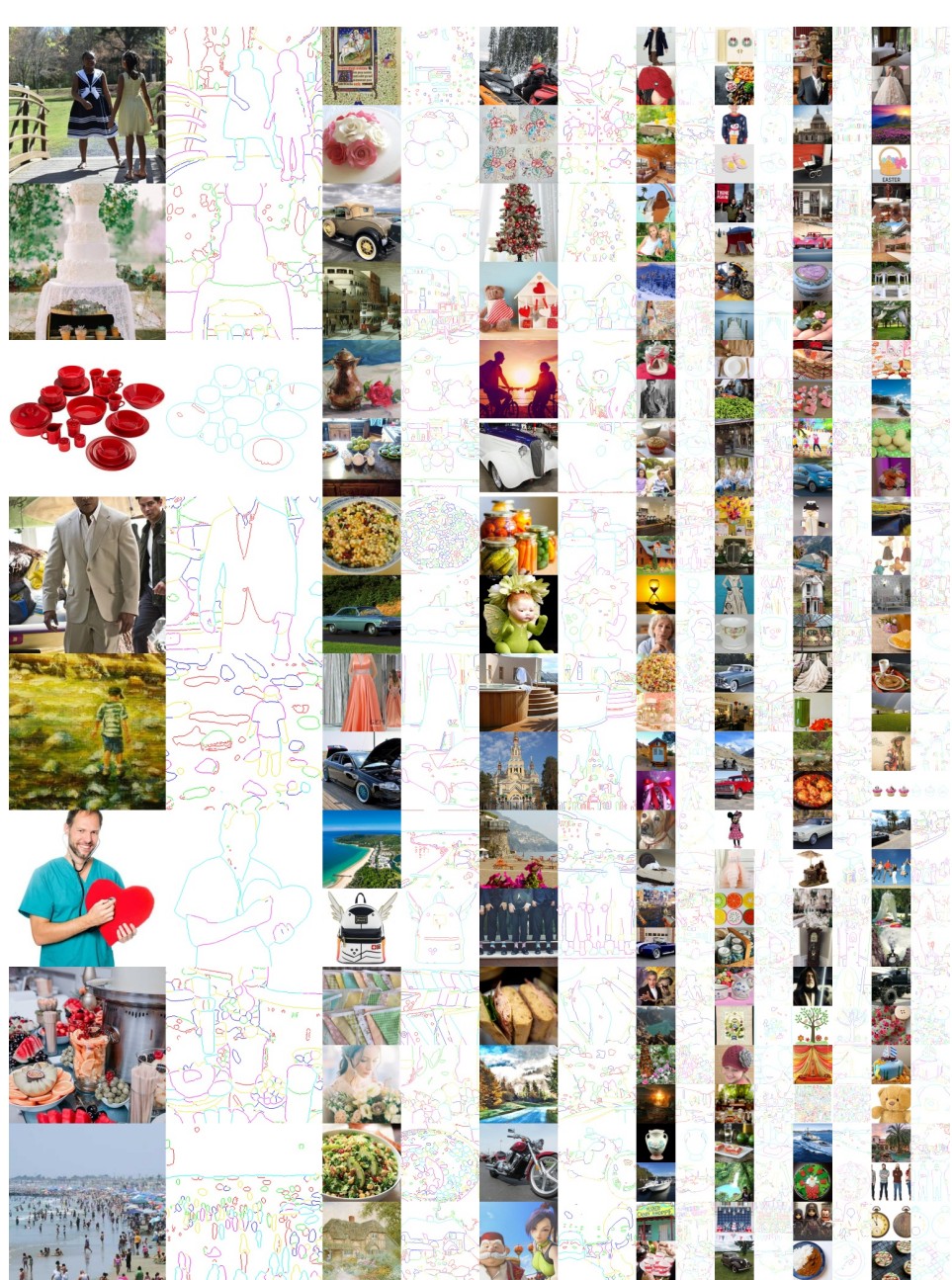

Figure 14: Synthetic Granular Edge Dataset samples. Pleas zoom in for more details. The edge map is shown with 6 granularity levels.

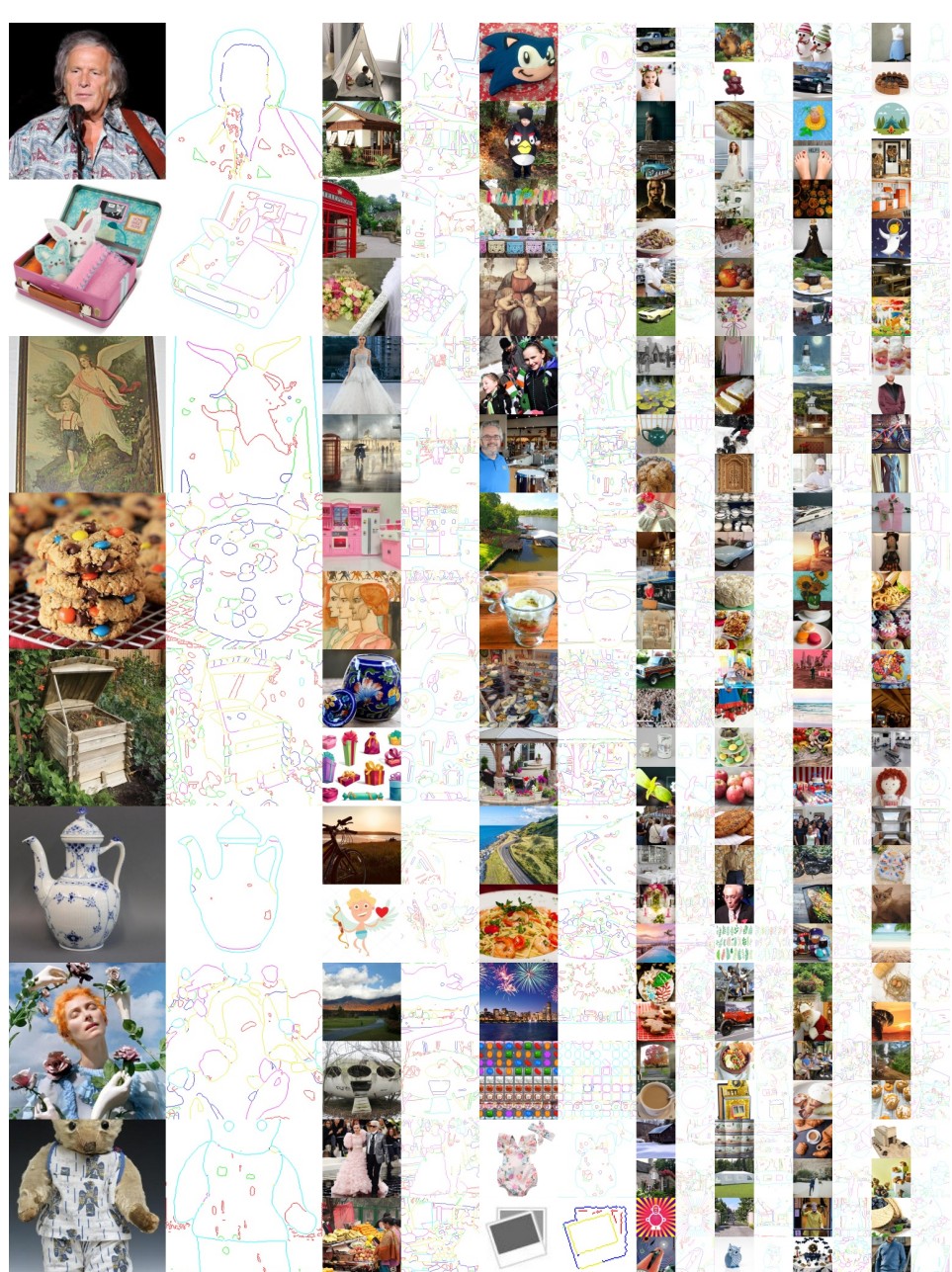

Figure 15: Synthetic Granular Edge Dataset samples. Pleas zoom in for more details. The edge map is shown with 6 granularity levels.

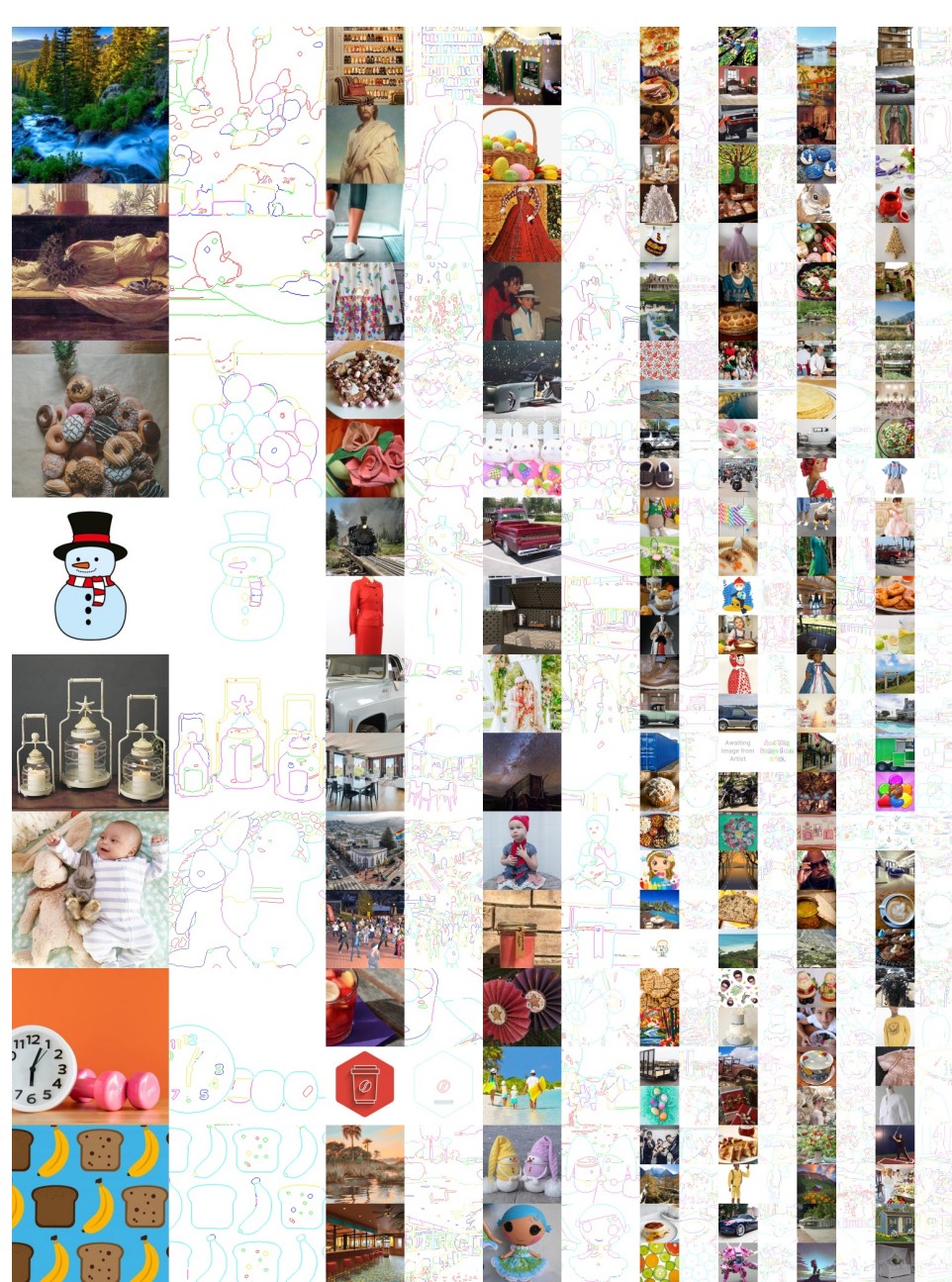

Figure 16: Synthetic Granular Edge Dataset samples. Pleas zoom in for more details. The edge map is shown with 6 granularity levels.

