# OpenReview forum: "Generalizable and Consistent Granular Edge Prediction"
_ICLR.cc/2026/Conference — Submitted to ICLR 2026_

### Official Review · Reviewer_8iwt · 2025-10-25

**Soundness:** 3
**Presentation:** 3
**Contribution:** 1
**Rating:** 4
**Confidence:** 2

**Summary:**

This paper constructs a synthetic dataset for granular  edge detection and proposes an efficient model that outputs edge maps of multiple granularity levels in a single forward pass, treating the task as a multi-class prediction problem. In addition, the paper introduces an edge consensus loss to enforce granularity coherence within edges, as well as  granularity-aware edge evaluation to demonstrate the effectiveness of the proposed approach.

**Strengths:**

1. The dataset is built by first extracting object boundaries using SAM and then performing refinement via graph-based representation, resulting in a large-scale granularity-aware edge detection dataset.
2. The proposed formulation effectively avoids the need for multiple forward passes required by previous methods to infer different granularity levels. By converting the problem into a multi-level classification task, the method can produce all granularity results with a single inference.
3. A new loss function and evaluation metric are introduced, which can potentially promote further research in granularity-aware edge detection.

**Weaknesses:**

1. Choice of granularity levels. The dataset is initially annotated with 36 granularity levels, but experiments later quantize these into 4 or 6 levels due to distribution imbalance (line 325–327). (1) How were the *original 36 levels* chosen? What advantage does the approach of annotating 36 levels *first and then merging* have over *directly annotating fewer granularity levels*? (2) When the granularity level is extremely small (level = 1), how does this differ from conventional deterministic edge detection? (3) In Fig. 12, one would expect that the smallest granularity level preserves only instance-related boundaries without background clutter. however, the results still appear influenced by background textures. Could the authors comment on this?
2. Model architecture clarification. The main paper does not clearly describe the model architecture. It only becomes clear in Appendix Table 3 that it follows a diffusion-model U-Net backbone. Were pretrained diffusion model weights used?
3. Inconsistent cross-dataset generalization. In Table 1, MuGE trained on SGED performs worse on NYUD compared to zero-shot inference from models trained on BSDS. However, performance improves on BIPED and Multicue. (1) Does this suggest that SGED yields weaker cross-domain generalization than BSDS? (2) Similarly, in Table 7, the performance gap between naive SAM and GEPM is small on BSDS and NYUD but large on BIPED and Multicue. Why does SGED training benefit some datasets but not others? (3) How does the current SOTA DiffusionEdge model perform when trained on SGED and tested on BSDS?
4. Model size comparison. In Table 7, what architecture does *naive SAM* refer to? Is it SAM-ViT-B? The base model used in the paper contains substantially more parameters than both SAM-ViT-B and the EfficientNet models used in MuGE. A direct parameter comparison table would clarify fairness in evaluation.
5. Class imbalance handling (Eq. 2). Since larger granularity levels have fewer annotated samples, does Eq. 2 account for this imbalance? Would class-balanced weighting or focal-type reweighting help further alleviate the imbalance noted in line 325–327?

**Questions:**

The strength of this paper lies in proposing a dataset, introducing a granularity consistency loss, and transforming prediction tasks of different granularities into multi-class classification tasks.

The main issue, however, lies in the experiments. The results on BSDS are nearly comparable to those of SAM's zero-shot performance. Additionally, the constructed dataset was expected to be entirely at the instance level when the granularity is set to 0, but this requirement has not been fully met.

---

> ### Author Response · Authors · 2025-11-23
>
> We sincerely thank the reviewer for the thoughtful feedback and suggestions. We have updated the manuscript with all newly added content marked in blue.
>
> **Reason to choose more granularity levels first and then merging.**
>
> We appreciate the reviewer for raising this question. There are several reasons to obtain a higher number of granularities first. 1) Granular edge prediction is a novel task. Annotating more granularities allows us to investigate how varying levels of granularity affect performance across datasets and tasks. 2) The synthetic annotation pipeline includes several parameters that influence edge density. If we were to annotate only a small number of levels directly, results would become overly sensitive to these parameter choices. In contrast, collecting richer edge maps first and then performing quantization allows for more stable statistics, better capturing the likelihood and saliency of an edge being perceived.
>
>
> **The granularity level GraLvl=1 case.**
>
> Setting the granularity level to 1 reduces the problem to traditional binary edge detection, as we noted in Section D.
>
> **Regarding the SGED in the coarsest granularity level.**
>
> We thank the reviewer for raising this question. We would like to emphasize that our dataset is entirely synthetic, and as such, it is not feasible to impose strong semantic constraints without extensive human supervision or prior knowledge, such as enforcing purely instance-level segmentation at the coarsest granularity. Achieving such semantics would require significant human annotation effort, which contradicts the goal of scalable, automatic data construction.
>
> Nevertheless, we have carefully selected the hyperparameters of SAM to ensure that, in most cases, the coarsest granularity predominantly captures object-level boundaries. While it is theoretically possible to enforce instance-level consistency by tuning parameters on a per-image basis, this would entail formidable manual effort and compromise scalability. Furthermore, it is practically impossible to define a fixed global parameter setting that satisfies this requirement across diverse images. A parameter that successfully filters out background clutter in one case may entirely eliminate valid object boundaries in another.
>
> **Model architecture clarification.**
>
> We appreciate the reviewer for pointing out the insufficient clarification in this part. While the architecture of GEPM is implemented using the “diffusers” package, we emphasize that the model is not a diffusion model. We only adopt the U-Net backbone component from diffusers.UNet2DModel for its simplicity, modularity, and ease of future reproducibility. This choice also facilitates public release and community usage by leveraging a widely supported and well-documented framework.
>
> Importantly, GEPM is trained entirely from scratch. No pretrained weights from diffusion models are used during training. Our usage of the “diffusers” codebase is purely for constructing the architecture and does not involve any generative or diffusion-based training objective. The full architecture can be exactly reconstructed by instantiating diffusers.UNet2DModel (https://github.com/huggingface/diffusers/blob/v0.35.1/src/diffusers/models/unets/unet_2d.py) using the configuration we provide.
> We have clarified and updated the above information in Section B.
>
> **Regarding model size.**
>
> We use SAM-ViT-L (sam2-hiera-large) model in Table 7, which is also the model we used for generating the synthetic training data.
> To address concerns about model size, we have evaluated the scalability of GEPM in Section B. In this experiment, we use a compact version of GEPM with only 94M parameters, comparable in size to the EfficientNet backbones used in MuGE. The results show that while there is a slight drop in edge consistency, the edge detection performance remains strong, demonstrating that our method is scalable to smaller architectures.

---

> > ### Author Response · Authors · 2025-11-23
> >
> > **Regarding questions in cross-dataset generalization and the choice of SOTA methods for granular edge training.**
> >
> > We appreciate the reviewer for providing the following three questions.
> >
> > Q1. SGED’s cross-domain generalization
> >
> > Q2. The reason for SGED benefiting more on BIPED and Multicue
> >
> > Q3. The reason to choose UAED and MuGE for SOTA for granular training on SGED but not DiffusionEdge.
> >
> > A1. We respectfully argue that it is not meaningful to assess cross-dataset generalization without considering the inherent annotation characteristics of each dataset. BSDS and NYUD are sparsely annotated, primarily focusing on salient object boundaries, whereas BIPED and SGED are densely annotated, capturing both prominent and subtle edges. As a result, BSDS-trained models naturally have better generalization to NYUD due to their shared perceptual similarity. In contrast, SGED, designed to provide rich annotations across granularity levels, aligns more closely with densely annotated datasets like BIPED and Multicue. This explains why SGED-trained models usually achieve better generalization on these datasets.
> >
> > A2. We acknowledge the observation noted by the reviewer and have provided a detailed discussion in Section F (lines 955–970). Importantly, training on SGED consistently improves performance across all datasets, although the magnitude of improvement varies. SAM, as a foundation model for object-level segmentation, is inherently aligned with datasets like BSDS and NYUD, which emphasize object boundaries. However, SAM performs less favorably on datasets with more diverse edge annotations, such as BIPED and Multicue, where GEPM offers significantly better predictions due to its granularity-aware training as evident in Table 8 and Figure 12. Moreover, we argue that the performance of granular comparison should also be considered. As shown in Table 9, GEPM shows substantial improvement in “granularity-aware edge evaluation” on the BSDS dataset.
> >
> > A3. We chose to adapt SOTA methods UAED and MuGE to SGED because both methods can be readily extended to handle granular edge prediction. In contrast, DiffusionEdge poses challenges in this context. Its architecture relies on a VAE decoder that produces continuous outputs in the [0,1] range. For binary edge detection, these values can be interpreted as edge probabilities and used with binary cross-entropy. However, granular edge prediction requires discrete class predictions compatible with multi-class cross-entropy, which the current DiffusionEdge architecture is not designed to support. Adapting it to the granularity prediction task would require substantial architectural modifications and require further investigation to find the proper way for adaptation.
> >
> >
> > **Regarding the class imbalance handling.**
> >
> > The primary motivation behind Equation 2 is to balance the contribution of edge and non-edge (background) regions in the loss, ensuring that the loss is evenly contributed from edge and background and the model is not overwhelmed by the majority background class. This edge-vs-background balancing strategy has been widely adopted and empirically validated across numerous deep learning-based edge detection works.
> >
> > Regarding class balancing by increasing the loss weights of each individual granular category, we did explore an alternative where each edge and background class receives equal loss contribution. However, we observed that this led to overly aggressive edge detection because of excessive loss weights from edges. Even subtle texture variations are falsely recognized as edges, particularly in setups with finer discretization (e.g., many granularity levels).
> >
> > We have also tried to have class-balanced weighting only for edge categories (i.e., after balancing edge/background, apply additional balancing within different edge granularity classes). However, this modification does not bring numeric significant change because we found the granularity class distribution in the SGED are relatively uniform, causing the impact of minor class imbalance inherently limited. Based on these findings, we adopt the formulation in Equation 2 as a more stable and empirically validated choice for training.

---

> > > ### Comment · Reviewer_8iwt · 2025-11-27
> > >
> > > Thank you for the response. Some of my concerns were addressed, but I still have a few remaining questions for clarification:
> > >
> > > 1. Why does increasing model size improve edge consistency but have limited impact on overall edge detection performance?
> > >
> > > 2. In Table 1, MuGE trained on SGED performs worse on NYUD compared to zero-shot models trained on BSDS, yet performs better on BIPED and Multicue. Could the authors clarify the main reason behind this asymmetry?
> > >
> > > 3. The motivation for “many levels first, then merging” is understandable, but why specifically choose 36? Can the authors demonstrate that 36 is a stable or optimal choice, and that using fewer levels would lead to the mentioned instability?

---

> ### Author Response · Authors · 2025-11-27
>
> We appreciate the reviewer for reading our response and would like to provide further clarification on the follow up question.
>
> **Why does increasing model size improve edge consistency but have limited impact on overall edge detection performance?**
>
> First, we would like to emphasize that GEPM’s edge detection performance is not universally limited when using increased model size. As noted in Ln 736-738 (line numbers are updated to the latest manuscript), the reported performance plateau pertains only to traditional binary edge evaluation protocol, which is a zero-shot task for GEPM. In contrast, under granularity-aware evaluation (as shown in Section 4.6 and Figure 7), GEPM exhibits consistent performance gains with larger models.
>
> Importantly, GEPM is trained on granular edge prediction, not binary classification. Thus, assessing its zero-shot generalization to binary tasks should consider not just model size, but also task alignment. While increasing model capacity improves tasks that the model is explicitly optimized for, such as edge consistency and granularity accuracy, we also observe modest improvements in binary edge detection. However, in this zero-shot setting, model size is not the dominant factor, and its impact remains limited compared to factors like task similarity and alignment between training and evaluation domains. Therefore, it can be inappropriate to attribute performance gains (or lack thereof) solely to model size without considering these underlying differences in task structure and domain alignment.
>
> **The reason for BSDS trained model has higher performance than SGED trained model on NYUD**
>
> We appreciate the reviewer’s observation and would like to kindly note that we have discussed this point in the initial rebuttal (A1 under the cross-dataset generalization section). To briefly summarize the core rationale:
>
> BSDS and NYUD are both sparsely annotated datasets, with an emphasis on salient object boundaries. In contrast, BIPED/Multicue and SGED are densely annotated, capturing not only prominent contours but also finer and less salient edges. Because of this annotation style, models trained on BSDS generalize better to NYUD due to their perceptual and semantic similarity. On the other hand, SGED, designed with dense and multi-level granularity annotations, aligns more closely with datasets like BIPED and Multicue, which leads to improved generalization on those benchmarks.
>
> Furthermore, we believe a direct comparison between “BSDS’s generalization on NYUD” and “SGED’s generalization on NYUD” can be misleading. There are two important factors at play:
> 1) Evaluation mismatch: Models trained on BSDS are optimized for binary edge detection, which is also the evaluation protocol used on NYUD. This constitutes a cross-dataset but not cross-task setting. In contrast, SGED-trained models are trained for granular edge prediction, and thus the binary evaluation on NYUD represents a cross-domain and cross-task scenario, making the generalization inherently more challenging.
> 2) Annotation source and quality: SGED is a synthetic dataset, whereas BSDS is manually annotated by humans. Human annotations often capture perceptually salient contours more reliably and with fewer artifacts, which may give BSDS-trained models a natural advantage when evaluated on object-focused datasets like NYUD.
>
> **Reason to have 36 granularity levels**
>
> We appreciate the reviewer’s thoughtful question. To clarify, we did not begin by choosing 36 granularity levels and then searching for hyperparameters. Instead, the process was the reverse: we first identified the SAM hyperparameters that meaningfully influence synthetic edge generation (Line 951), namely points_per_side, crop_n_layer, and stability_score_thresh. The first two use SAM’s recommended default candidates, three values for points_per_side and two for crop_n_layer. For stability_score_thresh, we empirically determined a meaningful operating range [0.85, 0.96]: lowering the threshold below 0.85 often fails to yield additional edges, while raising it above 0.96 often results in empty outputs. We then uniformly sampled six values within this range using a step of 0.02, as smaller intervals (e.g., 0.01) yielded highly similar edges that offered minimal additional diversity but significantly increased computational cost. Whereas larger steps can lead to abrupt changes in edge patterns, which would undermine the goal of producing a stable and smoothly varying granularity spectrum.
>
> Therefore, the resulting 36 levels arise naturally as the product of these choices: 3 (points_per_side) × 2 (crop_n_layer) × 6 (stability_score_thresh) = 36. This configuration ensures both sufficient granularity variation and dataset quality without unnecessary redundancy or instability.

---

### Official Review · Reviewer_YoKc · 2025-10-29

**Soundness:** 3
**Presentation:** 3
**Contribution:** 3
**Rating:** 6
**Confidence:** 3

**Summary:**

This paper introduces a new edge detection paradigm, termed Granular Edge Prediction (GEP), which redefines edge detection from binary classification into a granularity-aware prediction problem. Instead of simply detecting whether a pixel belongs to an edge, the proposed framework predicts an edge granularity level that reflects edge consistency and perceptual thickness.

To support this task, the authors construct a large-scale synthetic dataset called SGED, generated via SAM 2 segment masks and multi-level granularity transformations. They also design a novel Generalizable Edge Prediction Model (GEPM) equipped with a graph-based edge representation ensuring per-edge consistency, and an Edge Consensus Loss that enforces distributional agreement along the same edge.

**Strengths:**

1. Results show that GEPM achieves near or better than supervised methods in zero-shot settings on several benchmarks.

2. The Edge Consensus Loss based on Jensen–Shannon divergence effectively enforces intra-edge consistency in predictions.

**Weaknesses:**

1.  SGED is entirely synthetic; the paper lacks rigorous validation of how well its granularity annotations align with human perceptual judgments.

2. The distinction between “granular consistency” and previously studied “multi-level edge fusion” (e.g., in UAED or MuGE) could be more clearly articulated.

3. The quality of the SGED dataset depends heavily on SAM’s segmentation quality, which may introduce structured bias.

**Questions:**

1. How closely do the generated granularity annotations approximate human perceptual edge thickness? Have user or psychophysical studies been considered?

2. Could SGED overfit to SAM’s segmentation priors, limiting real-world generalization?

3. What is the training cost (GPU hours, parameter count), and how scalable is the proposed GEPM framework?

4. Would combining GEP with downstream tasks (e.g., boundary-aware segmentation) improve overall performance?

5. How sensitive is the model to the granularity level discretization (e.g., 36 vs. 10 levels)?

---

> ### Author Response · Authors · 2025-11-23
>
> We sincerely thank the reviewer for the thoughtful feedback and suggestions. We have updated the manuscript with all newly added content marked in blue.
>
> **The alignment between SGED’s granularity annotation and human perceptual judgment.**
>
> While it is infeasible to manually annotate all LAION images with perceptual granularity, we do evaluate the alignment between SGED’s granularity annotations and human perception in Section F of the supplementary material. Specifically, we apply the SGED generation pipeline to the BSDS dataset and compare the resulting predictions with human-annotated edges. As reported in Table 8 "Naive SAM", under a 6-level granularity setting, 29% of the predicted edge granularities exactly match the human-annotated levels, and over 2/3 fall within a 1-2 level difference. Given that SGED is fully synthetic and not explicitly calibrated using human labels, this level of alignment demonstrates its reasonable consistency with human perceptual judgments.
>
> **The distinction between “granular consistency” and previously studied “multi-level edge fusion” (e.g., in UAED or MuGE) could be more clearly articulated.**
>
> We appreciate the reviewer’s comment and would welcome further clarification regarding the term “multi-level edge fusion,” as this terminology does not appear explicitly in UAED or MuGE. A related concept in these works is “multi-granularity (or multi-level) edge prediction,” which refers to producing multiple binary edge maps at varying levels of granularity.
>
> In contrast, our notion of “granular consistency” is fundamentally different: it focuses on the intra-edge coherence of granularity assignments. That is, once an edge is identified, we assess whether the individual pixels belonging to that edge are predicted to have consistent granularity values. This notion captures the internal smoothness or semantic uniformity of an edge’s granularity, which is not addressed by prior multi-level prediction approaches.
>
> **Regarding SGED’s structural bias of segmentation prior.**
>
> We appreciate the reviewer for raising this question. While it is true that the SGED dataset may underrepresent certain non-object edges (e.g., textures, material boundaries, or contours not in segmentation prior), we observed that the GEPM model, once trained on SGED, generalizes well beyond the limitations of its synthetic training data. As illustrated in Figure 12 (in the updated manuscript), the BIPED dataset contains a large number of non-object edges that the original SAM model, which was used to generate SGED, fails to detect. In contrast, GEPM is able to accurately capture these edges despite having never seen human-annotated non-object boundaries during training. This generalization explains GEPM’s significant improvement over naive SAM on densely annotated datasets like BIPED and Multicue. A more detailed analysis of this behavior is provided in Ln 1017-1032 (line numbers are updated to the lastest manuscript) of Section F.
>
> **Training cost and scalability of GEPM**
>
> We have conducted a scalability analysis in Section B (see Table 4), where we evaluate models of varying sizes. The results show that while increasing model size leads to modest gains in edge detection accuracy, it brings more substantial improvements in edge consistency.
>
> Regarding training cost, training the “base” GEPM model requires approximately 700 V100 GPU hours, which translates to roughly 4 days on a machine with 8 V100 GPUs.

---

> > ### Author Response · Authors · 2025-11-23
> >
> > **GEPM’s potential for downstream tasks**
> >
> > We thank the reviewer for the valuable comment regarding the use of granular predictions in downstream tasks. To validate the utility of GEPM's granular outputs, we conducted a depth estimation experiment based on the publicly available implementation of BTS~\citep{bts}.\footnote{\url{https://github.com/cleinc/bts}} This method can be trained on a single dataset, making it a suitable choice to isolate the impact of adding granular edge priors without interference from mixed training sources.
> >
> > In our experiment, we train the BTS model on the NYUD depth estimation task. We use the GEPM model with GraLvl=6 to produce zero-shot granular edge predictions for NYUD images, as this configuration yields the best edge detection performance on NYUD as shown in Table 1. To inject the edge information into the BTS network, we concatenate the predicted granular edge map to each of the last four DenseNet feature stages of BTS after downscaling it using a lightweight 5-layer convolutional projection module, each with just 8 feature channels. This setup allows granular edge information to be integrated across multiple scales without significantly increasing model complexity.
> >
> > As shown in Table 7 (in the updated manuscript), injecting GEPM's granular predictions leads to modest but consistent improvements across most evaluation metrics, confirming the value of granularity-aware edge priors for downstream dense prediction tasks. A qualitative comparison in Figure 11 (in the updated manuscript) further demonstrates this benefit. By incorporating granular edges, the predicted depth map preserves sharper boundaries and clearer geometric structures, which are typically smoothed or lost in the baseline prediction.
> >
> > **Regarding the sensitivity to the granularity level discretization**
> >
> > The sensitivity of GEPM to granularity level discretization relates to two main factors: the number of granularity levels used and the characteristics of the evaluation dataset.
> >
> > Granularity level: As shown in Table 1, GEPM is relatively robust when granularity levels are set within a reasonable range (e.g., GraLvl = 2 to 15), with performance remaining stable across datasets. However, when the number of levels becomes extreme, the model becomes more dataset-dependently sensitive because of different dataset edge density.
> >
> > Evaluation dataset: Sensitivity also depends on the dataset’s inherent edge density. For example, when using GraLvl=1, GEPM underperforms on sparsely annotated datasets like BSDS and NYUD, because of over-dense predictions. In contrast, using GraLvl=36 leads to under-dense predictions that hurt performance on densely labeled datasets like BIPED and Multicue. This behavior stems from a mismatch between the predicted edge density and that of the ground-truth annotations, which becomes pronounced at extreme granularity settings.
> >
> > Overall, GEPM maintains consistent performance and is not sensitive to granularity level discretization under reasonable settings. The performance can be sensitive only at extreme granularity levels on certain datasets. These observations are further elaborated in Section D.

---

### Official Review · Reviewer_1wH2 · 2025-10-31

**Soundness:** 3
**Presentation:** 3
**Contribution:** 3
**Rating:** 6
**Confidence:** 4

**Summary:**

This work systematically defines and solves the "granular edge prediction" task for the first time, constructs the first large-scale, structured granular edge dataset (SGED) and a novel edge consensus loss and a comprehensive evaluation framework, providing a new paradigm for subsequent edge perception, controllable generation, and other tasks.

**Strengths:**

This work construct a large-scale synthetic dataset for granular edge prediction, where each edge is labeled with a quantized granularity level, and introduce a graphbased edge representation to enforce consistency in edge granularity across the dataset. The approach develop a novel edge consensus loss to enforce granularity consistency within individual edges, and propose a comprehensive evaluation framework, including granularity-aware edge evaluation and two quantitative metrics to assess the consistency of granular edge prediction.

**Weaknesses:**

1. The paper mentions "making it particularly valuable and has potential for applications where edge prominence varies," which shows that the authors recognize the importance of downstream applications. However, if the full text does not include specific experiments.
2. The SGED dataset relies on the Segment Anything Model (SAM) to generate synthetic edges. However, the core capability of SAM is to detect object boundaries, which leads to the dataset's insufficient capture of "non-object edges" (such as textures, contour boundaries, and edges with severe material differences).

**Questions:**

The core value of granular prediction is to support downstream tasks. Have the authors tried applying the granular output of GEPM to depth estimation or artistic rendering? Are there any empirical results demonstrating "performance improvement in downstream tasks "?

---

> ### Author Response · Authors · 2025-11-23
>
> We sincerely thank the reviewer for the thoughtful feedback and suggestions. We have updated the manuscript with all newly added content marked in blue.
>
> **GEPM’s potential for downstream tasks**
>
> We thank the reviewer for the valuable comment regarding the use of granular predictions in downstream tasks. To validate the utility of GEPM's granular outputs, we conducted a depth estimation experiment based on the publicly available implementation of BTS~\citep{bts}.\footnote{\url{https://github.com/cleinc/bts}} This method can be trained on a single dataset, making it a suitable choice to isolate the impact of adding granular edge priors without interference from mixed training sources.
> In our experiment, we train the BTS model on the NYUD depth estimation task. We use the GEPM model with GraLvl=6 to produce zero-shot granular edge predictions for NYUD images, as this configuration yields the best edge detection performance on NYUD as shown in Table 1. To inject the edge information into the BTS network, we concatenate the predicted granular edge map to each of the last four DenseNet feature stages of BTS after downscaling it using a lightweight 5-layer convolutional projection module, each with just 8 feature channels. This setup allows granular edge information to be integrated across multiple scales without significantly increasing model complexity.
>
> As shown in Table 7 (in the updated manuscript), injecting GEPM's granular predictions leads to modest but consistent improvements across most evaluation metrics, confirming the value of granularity-aware edge priors for downstream dense prediction tasks. A qualitative comparison in Figure 11 (in the updated manuscript) further demonstrates this benefit. By incorporating granular edges, the predicted depth map preserves sharper boundaries and clearer geometric structures, which are typically smoothed or lost in the baseline prediction.
>
> **The insufficiency of SGED in capturing “non-object edges”.**
>
> We appreciate the reviewer for raising this key observation. While it is true that the SGED dataset may underrepresent certain non-object edges (e.g., textures, material boundaries, or contours as mentioned), we observed that the GEPM model, once trained on SGED, generalizes well beyond the limitations of its synthetic training data. As illustrated in Figure 12 (in the updated manuscript), the BIPED dataset contains a large number of non-object edges that the original SAM model, which was used to generate SGED, fails to detect. In contrast, GEPM is able to accurately capture these edges despite having never seen human-annotated non-object boundaries during training. This generalization explains GEPM’s significant improvement over naive SAM on densely annotated datasets like BIPED and Multicue. A more detailed analysis of this behavior is provided in Ln 1017-1032 (line numbers are updated to the latest manuscript) of Section F.

---

> > ### Comment · Reviewer_1wH2 · 2025-11-23
> >
> > I appreciate the authors’ thorough and thoughtful response to my concerns. The additional depth estimation experiment with BTS on NYUD is well-designed and effectively isolates the contribution of GEPM’s granular edge priors. The supplementary evidence—especially GEPM’s strong performance on non-object edges in BIPED despite SGED’s object-centric construction—convincingly addresses my concern about the representational scope of the training data, highlighting the model’s capacity for compositional edge generalization.

---

### Official Review · Reviewer_kSjt · 2025-11-10

**Soundness:** 3
**Presentation:** 3
**Contribution:** 3
**Rating:** 4
**Confidence:** 3

**Summary:**

This paper introduces Granular Edge Prediction, extending binary edge detection to predict granularity levels reflecting perceptual saliency. Main contributions: (1) SGED dataset with 376K SAM-generated images and graph-based refinement for consistency; (2) GEPM model with Edge Consensus Loss; (3) new consistency metrics. The method achieves competitive zero-shot performance across four benchmarks. However, the work lacks human validation, inherits SAM's biases toward object boundaries, and shows limited advantages over supervised methods.

**Strengths:**

1. The granular edge prediction task is well-motivated, addressing the inherent subjectivity in edge annotation with clear practical applications.
2. Creating 376K images addresses severe data scarcity (only 600 in existing datasets), enabling robust training and generalization.
3. Table 1 shows competitive cross-dataset results, with Multicue ODS of 0.843 approaching supervised methods (0.904).

**Weaknesses:**

1. Zero human studies validating predicted granularities align with perception.
2. Critical omission for a paper claiming to predict "human-recognized" edge granularity.
3. Some figures hard to see, granularity reduction (36→6) under-motivated in main text.

**Questions:**

1. Can you provide human studies validating predicted granularities?
2. What is supervised fine-tuning performance?
3. When is zero-shot granular prediction preferable to supervised binary detection?

---

> ### Author Response · Authors · 2025-11-23
>
> We sincerely thank the reviewer for the thoughtful feedback and suggestions. We have updated the manuscript with all newly added content marked in blue.
>
> Regarding the visibility of some figures, we would like to note that our figures are provided in high-resolution digital format. For optimal viewing, we recommend reading the paper in digital form and zooming in on the figures when necessary.
>
> **Granularity reduction**
>
> We reduced the number of granularity levels for two key reasons. First, having too many granularity classes dilutes the sample count per class, making individual edge categories sparse and increasing the dominance of the background class during training. Reducing the granularity levels helps mitigate this issue. Second, it improves tolerance to prediction noise in synthetic annotations: due to slight variations in edge prominence, similar edges may be labeled with adjacent granularity scores across samples. Without reduction, a minor prediction error could incur a large penalty. By quantizing the labels into coarser buckets, such minor discrepancies are absorbed within the same class, enhancing training robustness.
>
>
> **Alignment between granularity prediction and human perception.**
>
> We appreciate the reviewer’s insightful question regarding the validation of predicted granularities against human perception. We would like to clarify that Granularity-Aware Edge Evaluation (Section 4.6, Figure 4 & 7) is specifically designed to evaluate this alignment. Directly collecting human-annotated ground-truth granularity labels is inherently infeasible because human annotators typically only produce binary edge maps rather than explicitly rating edge salience or granularity. To assess alignment to human perception, our evaluation uses the frequency of edge labeling across multiple annotators in the dataset. Statistically, the frequency with which an edge is annotated reflects its perceptual salience. We compare this with the model-predicted granularity to assess alignment. A strong correspondence indicates that predicted granularities successfully capture perceptual edge importance. Notably, GEPM achieves the highest proportion of correctly matched granularities among all methods, with Figure 4 qualitatively demonstrating that GEPM’s predictions correlate well with human-labeled consensus. This evaluation framework provides a meaningful and statistically grounded proxy for validating perceptual granularity alignment, even in the absence of direct human granularity labels.
>
>
> **Supervised Tuning Performance**
>
> We conduct a feature probing experiment using a two-layer MLP on the “base” GEPM (GraLvl=6) model to assess its suitability for downstream tasks. Specifically, we replace the final layer with a binary prediction head (the two-layer MLP) and freeze all other pretrained layers. This setup is both parameter-efficient and evaluates the quality of learned features. The total number of trainable parameters is 20K (0.007% of the full model). The fine-tuned model improves ODS/OIS/AP by 4.35%/2.72%/2.34%, narrowing the gap to the SoTA (supervisedly trained on BSDS) by 60.4%/50.4%/31.2%, respectively. The corresponding results have been updated in the Sec D. We believe further gains are possible with more sophisticated training strategies.
>
> **When is zero-shot granular prediction preferable to supervised binary detection?**
>
> We thank the reviewer for the thoughtful question. Zero-shot granular edge prediction is particularly advantageous in two common scenarios:
>
> 1. Tasks requiring adaptive detail control. Granular Edge Prediction offers a unified framework that can flexibly adjust the level of edge detail without retraining. This is especially beneficial in applications where the required edge precision varies across scenes or tasks. For example, in industrial defect detection, different types of damage such as cracks, corrosion, or surface wear exhibit varying levels of visibility. With GEP, coarse granularity predictions can isolate prominent structures (e.g., clear fractures), while fine-granularity predictions help detect subtle or partially eroded regions, without requiring different models or manual hyper-parameter tuning.
>
> 2. Label-free deployment in new domains. In settings where no human-annotated edge maps are available (e.g., newly collected datasets or niche imaging modalities), supervised edge detection is not applicable. GEPM enables immediate deployment in such domains, offering rich edge predictions that include pixel-wise granularity scores. These can be thresholded to retain only the most salient edges or include more ambiguous contours, depending on user intent, thus allowing downstream users to control the abstraction level without retraining or collecting costly labels.
>
> We have elaborated the Ln 48-51 in the manuscript to clarify the advantage cases of zero-shot granular prediction.

---

### Author Response · Authors · 2025-12-02
**Submission 6772 Authors' Summary of Rebuttal**

We appreciate all reviewers and the AC for their efforts in reading our submission and providing insightful comments. After being notified of last week’s security incident of the conference, we, as a part of being directly affected, felt frustrated that the discussion phase could not proceed completely. Nonetheless, we are grateful to the committee for their efforts to recover from the situation. We understand that the reassignment of submissions imposes a significant workload on the AC, and we sincerely thank you for the time and consideration. Below, we provide a summary of our rebuttal.

Our work is a pioneering effort on a novel task, “Granular Edge Detection,” which extends traditional binary edge prediction to predicting the perceptual saliency of edges. This “**new edge detection paradigm**” (**YoKc**) has been described as “**well-motivated, addressing the inherent subjectivity in edge annotation with clear practical applications**” (**kSjt**) and as “**systematically defining and solving the “granular edge prediction” task for the first time**” (**1wH2**).

Our contributions have been widely recognized by the reviewers, including:
A large-scale synthetic granular dataset built with a novel graph-based edge representation. This dataset “**addresses severe data scarcity, enabling robust training and generalization**” (**KSjt, 1wH2, YoKc, 8iwt**)
A novel Edge Consensus Loss function with a solid mathematical derivation based on Jensen–Shannon divergence, which “**effectively enforces intra-edge consistency in predictions**” (**kSjt, 1wH2, YoKc, 8iwt**)
A comprehensive evaluation framework that “**can potentially promote further research in granularity-aware edge detection.**” (**1wH2, 8iwt**)
The proposed GEPM model achieved “**competitive cross-dataset results**” and “**near or better than supervised methods in zero-shot settings on several benchmarks.**” (**kSjt, YoKc**)

---

During the rebuttal, we actively engaged in discussion and sufficiently addressed reviewers’ concerns.

Reviewer **1wH2** and **YoKc** noted that our edge priors were derived from an instance segmentation model, raising concerns that the predictions might be biased toward object boundaries. After our clarification with Figure 12, reviewer **1wH2** acknowledged that this “**supplementary evidence convincingly addresses my concern about the representational scope of the training data, highlighting the model’s capacity for compositional edge generalization.**”

Reviewer **kSjt** questioned the performance of GEPM under supervised fine-tuning for binary edge detection, while **1wH2** and **YoKc** asked whether GEPM’s predictions improve downstream tasks. In response, we added experiments in Section D on supervised fine-tuning and a downstream task (depth estimation). These experiments sufficiently addressed these concerns and were recognized by **1wH2** as “**well-designed and effectively isolating the contribution of GEPM’s granular edge priors.**”

---

> ### Author Response · Authors · 2025-12-02
> **Submission 6772 Authors' Summary of Rebuttal (cont.)**
>
> Reviewer **kSjt** and **YoKc** asked about the alignment between the granularity annotation/prediction and human perception. We clarified in the rebuttal that such alignment is demonstrated in the “Granularity-Aware Edge Evaluation” assessment in the Section 4.6 and F.
>
> We addressed concerns from reviewer **YoKc** and **8iwt** about GEPM’s scalability and training cost. In the second round response to reviewer **8iwt**, we further clarified the effect of model size on both zero-shot binary evaluation and granular assessment.
>
> Across the two rounds of discussion with reviewer **8iwt**, we first addressed the concerns regarding the granularity design choice, the synthetic generation setting, cross-dataset generalization and class imbalance handling. After these concerns were recognized as "**addressed**"", we further provide, in the second round rebuttal, additional clarification on 1) the performance generalization of SGED-trained model (i.e., synthetic data trained model) and BSDS-trained model (i.e., real data trained model) and 2) the rationale for selecting 36 granularity levels. These updates are reflected in Section 4.3 and Section E.
>
> In discussion with reviewer **kSjt**, we elaborated our motivation for granularity reduction and updated the corresponding context in Section 4.3.
>
> We addressed the remaining concerns from reviewer **YoKc** by 1) clarifying the concept regarding granular consistency, and 2) providing a comprehensive discussion regarding the sensitivity to the granularity level discretization with the content of Table 1 and Section D.
>
> We further updated our manuscript to fully address reviewer **kSjt**’s question on the cases of preference of zero-shot granular prediction against supervised binary detection in Section 1, and reviewer **8iwt**’s questions on the details of the model architectures in Section B.
>
> We sincerely thank all reviewers for their valuable comments and have carefully revised the manuscript to address them. Finally, we truly appreciate the AC’s time and effort in reviewing our work and would respectfully plead that our discussions with reviewers be considered in the final decision.
>
> Sincerely,
> Authors of Submission 6772

---

### Meta-Review · Area_Chair_aMtu · 2025-12-12

**Summary:**

This paper presents a synthetic dataset for granular edge detection and proposes an efficient model that outputs edge maps of multiple granularity levels. Experiments on granularity-aware edge evaluation to demonstrate the effectiveness of the proposed approach. While the benchmark and model design are generally well-received, reviewers raise substantial concerns regarding the reliability of granularity labels, alignment with human perception, and the dependence on SAM-generated masks during synthetic data creation. This paper would benefit from stronger validation of predicted granularities against human labels.

**Reviewer Concerns:**

The major concerns focus on whether the predicted edge granularities truly correspond to human-recognized levels, given the lack of direct human-annotated granularity supervision. Reviewers also question potential over-claims about predicting “human-recognized” granularity, the limitations of relying on SAM for synthetic generation, the risk that the method may inherit SAM’s segmentation priors, the unclear selection and merging of granularity levels, and inconsistent generalization across datasets. While the authors’ rebuttal provides clarifications and partial evidence addressing several of these issues, concerns about human alignment, dataset generation assumptions, and evaluation reliability remain outstanding.

**Reviewer Scores:**

Reviewer kSjt gave an initial score of 4, with major concerns regarding evaluation against human annotators and over-claiming human-recognized granularities. The rebuttal partially addresses these issues, but alignment with human judgment remains unconvincing; the reviewer may keep the score at 4.
Reviewer 1wH2 gave an initial score of 6, focusing on limitations in the synthetic generation process—particularly SAM missing non-object edges—and questions about generalization to downstream tasks. Although most concerns were addressed, the reviewer does not indicate an intention to raise the score and may keep it at 6.
Reviewer YoKc gave an initial score of 6, highlighting concerns about evaluation quality, possible overfitting to SAM’s priors, and a mismatch between predicted and human-judged granularities under a 6-level scheme. Only part of these issues were resolved in the authors’ response, so the reviewer is likely to keep the score at 6.
Reviewer 8iwt gave an initial score of 4, citing unclear granularity-level choices and inconsistent cross-dataset generalization. The rebuttal clarified the rationale for granularity merging and explained the observed performance differences, and the reviewer may raise the score to 6.

---

### Decision · Program_Chairs · 2026-01-26

Reject